# Self-reactivity controls functional diversity of naive CD8+ T cells by co-opting tonic type I interferon

Young-Jun Ju[1,6], Sung-Woo Lee[2,3,4,6], Yoon-Chul Kye [1], Gil-Woo Lee[2,3,4], Hee-Ok Kim[4], Cheol-Heui Yun [1✉] & Jae-Ho Cho [3,4,5✉]

The strength of the T cell receptor interaction with self-ligands affects antigen-specific immune responses. However, the precise function and underlying mechanisms are unclear. Here, we demonstrate that naive CD8+ T cells with relatively high self-reactivity are phenotypically heterogeneous owing to varied responses to type I interferon, resulting in three distinct subsets, CD5loLy6C−, CD5hiLy6C−, and CD5hiLy6C+ cells. CD5hiLy6C+ cells differ from CD5loLy6C− and CD5hiLy6C− cells in terms of gene expression profiles and functional properties. Moreover, CD5hiLy6C+ cells demonstrate more extensive antigen-specific expansion upon viral infection, with enhanced differentiation into terminal effector cells and reduced memory cell generation. Such features of CD5hiLy6C+ cells are imprinted in a steady-state and type I interferon dependence is observed even for monoclonal CD8+ T cell populations. These findings demonstrate that self-reactivity controls the functional diversity of naive CD8+ T cells by co-opting tonic type I interferon signaling.

[1] Department of Agricultural Biotechnology and Research Institute of Agriculture and Life Sciences, Seoul National University, Seoul 08826, Korea. [2] Division of Integrative Biosciences and Biotechnology, Pohang University of Science and Technology, Pohang 37673, Korea. [3] Department of Microbiology and Immunology and Medical Research Center for Combinatorial Tumor Immunotherapy, Chonnam National University Medical School, Hwasun 58128, Korea. [4] Immunotherapy Innovation Center, Chonnam National University Medical School, Hwasun Hospital, Hwasun 58128, Korea. [5] BioMedical Sciences Graduate Program, Chonnam National University Medical School, Hwasun 58128, Korea. [6]These authors contributed equally: Young-Jun Ju, Sung-Woo Lee. ✉email: cyun@snu.ac.kr; jh_cho@chonnam.ac.kr

CD8+ T cells are heterogeneous in phenotype, function, and tissue distribution in mice and humans[1–3]. These characteristics are particularly well established in effector and memory populations, and intensive studies have been performed to elucidate the factors and mechanisms responsible for such heterogeneity[4,5]. Unlike the antigen-primed cells, naive CD8+ T cells are typically considered as a homogenous population, and little attention has been paid to their complexities. However, recent studies have shown that such cells are not uniform and are separated into different subpopulations that exhibit different functional properties[6–10]. The strength of the T cell receptor (TCR) interaction with self-ligands is a major factor that can lead to such differences. As the strength is highly variable among individual TCRs and proportional to CD5 expression[11–13], many studies have focused on the comparison between CD5 low (CD5lo) and high (CD5hi) cells. In these studies, CD5hi cells demonstrated better functional responsiveness to various stimuli and more extensive antigen-specific expansion upon pathogen infection[7–9], supporting the current notion that the intrinsic self-reactivity determines the immune responses of naive CD8+ T cell pools.

The precise mechanisms underlying the differential responses between CD5lo and CD5hi cells are unclear. The coincidental positive correlation between TCR affinity to cognate foreign antigens and self-ligands appears to explain this phenomenon[8]. Thus, CD5hi cells with relatively high self-reactivity must always induce more antigen-specific immune responses than CD5lo cells[8]. Although the mechanism establishing this relationship has not been addressed, this perspective suggests that the immune system produces CD5lo cells unnecessarily and eventually eliminate them with the gradual accumulation of CD5hi cells[8]. Alternatively, different explanations have been proposed in other studies based on the results that CD5hi cells are inherently more functional than CD5lo cells[6,9]. Thus, CD5hi cells exhibit higher extracellular signal-regulated kinase activity and more IL-2 production upon stimulation with phorbol myristate acetate (PMA)[9] and higher proliferation rates upon stimulation with cytokines such as interleukin (IL)−2, IL-7, and IL-15 than CD5lo cells[6]. Enhanced responses of CD5hi cells have also been shown to be associated with increased mRNA expression of transcripts encoding genes associated with cell activation and division[7,10].

Despite the fact that self-reactivity differentially affects the inherent functional property and induces different immune responses of naive T cells, these issues are further complicated by recent findings demonstrating that CD5hi cells exhibit less responsiveness upon TCR ligation than CD5lo cells[14]. Although this result is consistent with the originally defined role of CD5 as a negative regulator of TCR signaling[13], the finding makes it difficult to understand the aforementioned positive effect of self-reactivity on T cell function. Furthermore, unlike CD5lo cells, CD5hi cells have a small portion of subpopulations that express Ly6C and CD183[7], although the functional roles of these subsets have not been determined. Therefore, different T cell immune responses may not result from the difference in TCR strength but instead reflect indirect roles of several other factors, such as physiologically relevant cytokines, in a steady state. Under this condition, CD5hi cells with relatively high self-reactivity have a greater advantage than CD5lo cells to respond to such cytokines[6,10,15]. Thus, the questions of which cytokines are involved, and how they affect the inherent function and fate of T cells need to be clarified for both the thymic and post-thymic stages of T cells.

Here, we show that relatively high self-reactivity facilitates phenotypic and functional diversity of naive CD8+ T cells by co-opting tonic type I interferon (IFN) signals at variable degrees. These findings provide insight into the mechanism of the coordination of T cell self-reactivity with physiological cues for shaping the functional fitness of naive CD8+ T cell pools, implicating type I IFN as a coordinating factor that contributes to complexity and diversity of adaptive T cell immune responses.

## Results

**CD5hi naive CD8+ T cells are composed of phenotypically heterogeneous subsets.** In naive CD8+ T cells, CD5hi cells were shown to exhibit more complex phenotypes and induce more antigen-specific immune responses than CD5lo cells[7]. To investigate the mechanisms underlying this phenomenon, we first compared phenotypic differences between CD5lo and CD5hi subsets of CD44lo naive CD8+ T cells from C57BL/6 (B6) mice. Among the various markers that were evaluated, Ly6C and CD183 (and to a lesser extent CD103) demonstrated the most remarkable differences (~2–6 times higher in CD5hi than in CD5lo cells; Supplementary Fig. 1a). Indeed, a small fraction of CD44lo naive cells exhibited relatively high Ly6C and CD183 expression but low CD103 expression, although this characteristic was typical in CD44hi memory-phenotype (MP) cells (Fig. 1a). Approximately all Ly6C+, CD183+, and CD103− subsets of CD44loCD8+ cells were derived from CD5hi cells (~90–95%; Fig. 1b). Despite the expression of such memory markers, these cells were still phenotypically and functionally naive, as evidenced by their much lower levels of CD44, CD122, Ly6C, and CD183 expressions, lower proliferation rates upon TCR (or cytokines, IL-7 and IL-15) stimulation, and lower IFN-γ producing abilities upon 5 h stimulation with PMA and ionomycin than those of MP cells (Supplementary Fig. 1b–d).

When analyzing the lineage relationship among Ly6C+, CD183+, and CD103− subsets in CD5hi cells, the majority of the CD183+ and CD103− subsets (~80–85%) were observed in Ly6C+ cells (Fig. 1c). Similarly, in the thymus, nearly all Ly6C+ cells were detected in CD5hi cells of mature CD24lo (but not immature CD24hi) CD4−CD8+ single-positive (SP) thymocytes (Fig. 1d). In these CD24lo SP thymocytes, CD103− cells were also observed in mainly CD5hi cells (~85–90%; Supplementary Fig. 1e). CD183+ cells, unlike Ly6C+ and CD103− cells, were rarely observed in the thymus (Fig. 1d), implying post-thymic generation.

Next, we examined the extent to which the peripheral Ly6C+ subset is derived from thymic counterparts. In neonatal and young adult B6 mice, the proportion of thymic Ly6C+ cells remained constant, comprising ~5-10% of total CD4−CD8+ SP thymocytes, until day 40 after birth (Fig. 1e, left). However, splenic Ly6C+ cells accounted for ~20% of total CD44lo CD8+ T cells immediately after birth, and this percentage increased to 30% by day 5, decreased slightly on day 7, and subsequently persisted at ~20–30% (Fig. 1e, left). Thus, splenic Ly6C+ cell numbers increased only slightly during the neonatal period (days 1–7). However, the amount increased significantly by day 40 (~6-fold higher than that of thymic Ly6C+ cells; Fig. 1e, right). An unexpectedly high frequency of splenic CD183+ cells (~40% of total naive CD8+ cells) was observed from day 1, with this percentage decreasing sharply on day 3 (~10–20%) and bottoming out on day 40 (<10%), although their cell numbers during this period gradually increased (Supplementary Fig. 1f). These data suggest that the peripheral naive Ly6C+ subset is derived from both the thymus and periphery (~33% and ~67%, respectively), with some CD183+ cells being derived in the periphery.

The proportion of the peripheral Ly6C+ (and CD183+) subset remained relatively constant over time (>1 yr) despite a reduction in the actual cell numbers in aged mice owing to thymic involution (Fig. 1f and Supplementary Fig. 1g). Notably, splenic

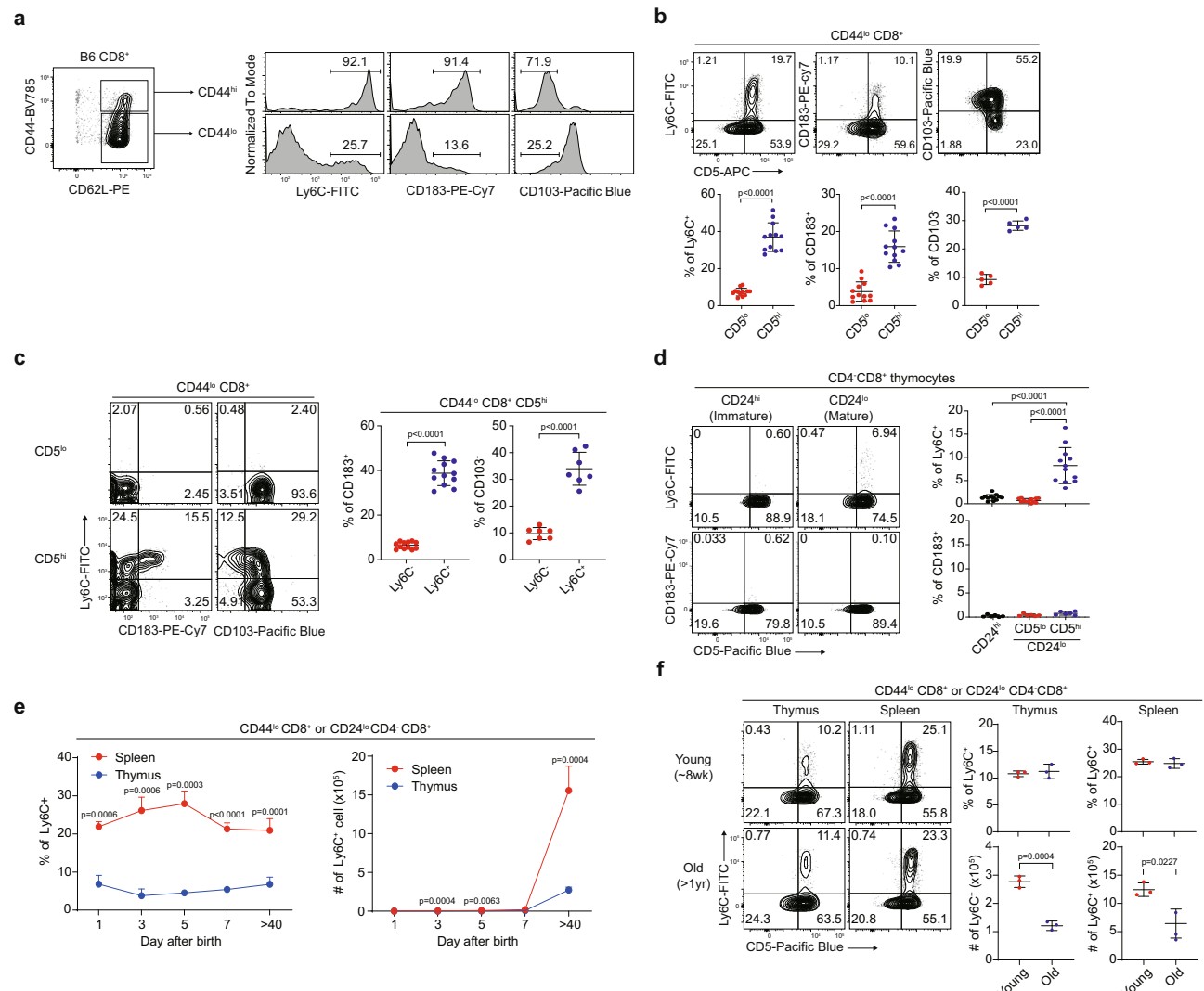

**Fig. 1 Peripheral naive CD8+ T cells are phenotypically heterogeneous. a** Ly6C, CD183, and CD103 expression on B6 CD44lo and CD44hi CD8+ T cells. **b** Percentage of Ly6C+, CD183+, and CD103− cells in CD5lo and CD5hi B6 naive CD8+ T cells ($n = 12$ mice for Ly6C+ and CD183+, $n = 5$ mice for CD103−). **c** Percentage of CD183+ and CD103− cells in Ly6C− and Ly6C+ CD5hi B6 naive CD8+ T cells ($n = 12$ mice for CD183+, $n = 7$ mice for CD103−). **d** Percentage of CD183+ and Ly6C+ cells in CD24hi, CD24lo CD5lo, and CD24lo CD5hi B6 CD4−CD8+ thymocytes ($n = 12$ mice for Ly6C+, $n = 6$ mice for CD183+). **e** Splenic and thymic Ly6C+ B6 naive CD8+ T cell changes over time after birth ($n = 3$ mice for each group). **f** Splenic and thymic Ly6C+ B6 naive CD8+ T cells in young and old mice ($n = 3$ mice for each group). Statistical significance was confirmed by two-tailed unpaired Student's $t$-test. Results are shown as mean ± SD. Data representative of 2–3 independent experiments. Source data are provided as a Source data file.

naive Ly6C+CD5hi cells demonstrated no apparent bias in the diversity of TCR α- and β-chains when compared to that in the Ly6C−CD5hi cells (and even in Ly6C−CD5lo cells; Supplementary Fig. 1h). Together, these results indicate that peripheral naive CD8+ T cells are phenotypically heterogeneous, particularly with respect to CD5hi cells, and are primarily divided into three distinct subsets, CD5loLy6C−, CD5hiLy6C−, and CD5hiLy6C+ cells (hereafter referred to as CD5lo, Ly6C−, and Ly6C+ cells, respectively).

**Naive CD8+ T cells are continuously exposed to type I IFN in a steady state**. Given that the peripheral naive Ly6C+ cells are subjected to stable maintenance, we investigated the factor responsible for their homeostasis. Type I IFN was demonstrated to affect Ly6C expression in CD4+ and CD8+ T cell populations[16,17]. In fact, the percentage and number of Ly6C+ naive CD8+ T cells were significantly lower in mice lacking the

receptor for either type I IFN (Ifnar−/−) or both type I IFN and IFN-γ (Ifnar−/−.Ifngr−/−) and in mice lacking STAT1 (Stat1−/−), a crucial component of IFN signaling, than in wild-type (WT) mice (Fig. 2a). This reduction was observed in the spleen and even in the thymus (~4–5- and 20-fold reductions, respectively; Fig. 2b). Unlike Ly6C+ cells, the CD183+ cell percentage and number remained unchanged in these mice (Supplementary Fig. 2a). Type I IFN signaling was also required for Ly6C+ cell generation in neonatal mice (Supplementary Fig. 2b), demonstrating continuous exposure to type I IFN throughout their lifetime. Moreover, in vitro culture with IFN-β but not with other cytokines (IFN-γ, IL-2, IL-4, IL-6, IL-7, IL-12, IL-15, IL-21, IL-23, and TGF-β) induced Ly6C expression from purified Ly6C− naive CD8+ T cells (Fig. 2c and Supplementary Fig. 2c). Furthermore, a significant fraction of WT, but not Ifnar−/−, Ly6C− naive CD8+ T cells was able to convert to Ly6C+ cells on day 7 after adoptive transfer to B6 hosts (Fig. 2d). In contrast, CD183+ cells, albeit at lower levels, were similarly observed in both WT and Ifnar−/−

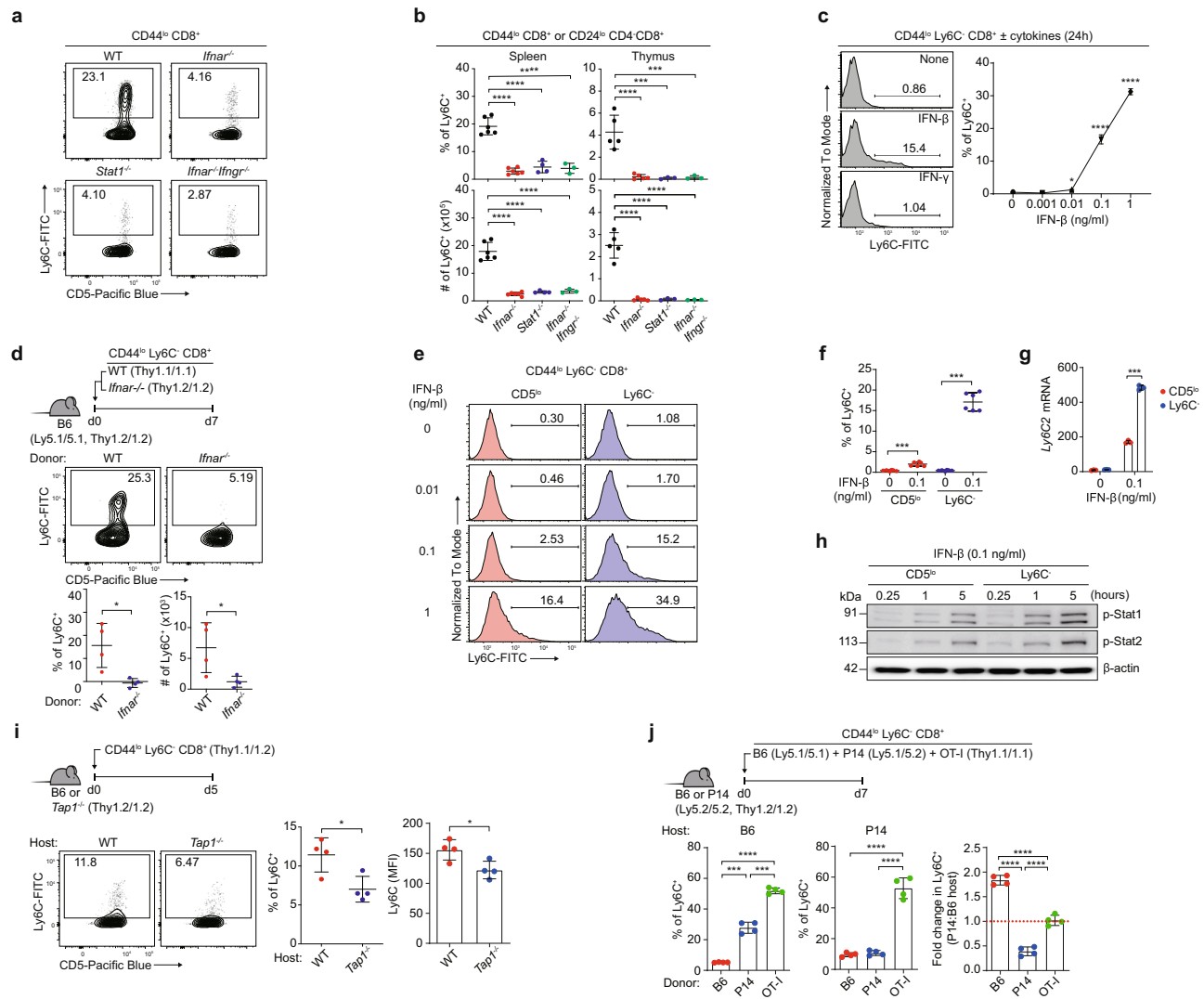

**Fig. 2 Type I IFN and self-TCR are crucial for inducing Ly6C⁺ naive CD8⁺ T cells. a** Percentage of B6 Ly6C⁺ naive CD8⁺ T cells in spleen from different type I IFN signal deficient mice. **b** Percentage and absolute number of B6 Ly6C⁺ naive CD8⁺ T cells in spleen and thymus from different type I IFN signal deficient mice ($n = 6$ for WT, $n = 6$ for $Ifnar^{-/-}$, $n = 4$ for $Stat1^{-/-}$, $n = 3$ for $Ifnar^{-/-}$ $Ifngr^{-/-}$ mice). **c** Percentage of induced Ly6C⁺ cells from B6 Ly6C⁻ naive CD8⁺ T cells in vitro ($n = 3$ mice for each group). **d** Percentage and absolute number of induced Ly6C⁺ cells from WT and $Ifnar^{-/-}$ B6 Ly6C⁻ naive CD8⁺ T cells in vivo ($n = 4$ mice for each group). **e, f** Percentage of induced Ly6C⁺ cells from CD5ˡᵒ and Ly6C⁻ B6 naive CD8⁺ T cells cultured with IFN-β ($n = 6$ mice for each group). **g** mRNA expression of $Ly6c1$ from CD5ˡᵒ and Ly6C⁻ B6 naive CD8⁺ T cells cultured with IFN-β ($n=3$ mice for each group). **h** Phosphorylation of STAT1 and STAT2 in CD5ˡᵒ and Ly6C⁻ cells cultured with IFN-β. **i** Percentage of induced Ly6C⁺ cells and Ly6C MFI from B6 Ly6C⁻ naive CD8⁺ T cells transferred to WT or $Tap1^{-/-}$ hosts ($n = 4$ mice for each group). **j** Percentage of induced Ly6C⁺ cells from B6, P14, or OT-I Ly6C⁻ naive CD8⁺ T cells transferred to B6 or P14 hosts ($n = 4$ mice for each group). Statistical significance was confirmed by two-tailed unpaired Student's $t$-test. Results are shown as mean ± SD. Data representative of 2-3 independent experiments. Source data are provided as a Source data file.

donor cells, with both donors still remaining CD44ˡᵒ naive phenotype (Supplementary Fig. 2d, e).

**Type I IFN-induced Ly6C in naive CD8⁺ T cells depends on TCR self-interactions**. The reason for the selective influence of type I IFN on Ly6C⁺ cell generation, which was restricted to CD5ʰⁱ and not CD5ˡᵒ cells, was unclear. Given the known correlation between self-reactivity and cytokine responsiveness[6], we postulated that two parameters were involved, namely, the strength of TCR self-interaction and relative sensitivity to type I IFN. To investigate this, purified Ly6C⁻ CD5ˡᵒ and CD5ʰⁱ naive CD8⁺ T cells were cultured in the presence of varying doses of IFN-β (Fig. 2e–h and Supplementary Fig. 2f). CD5ʰⁱ cells showed much higher sensitivity to type I IFN than CD5ˡᵒ cells, leading to the induction of Ly6C expression at both the protein and mRNA

levels (Fig. 2e–g). This disparate sensitivity was not attributed to the different levels of IFNAR (IFNAR1 and IFNAR2; Supplementary Fig. 2g) but rather to the variation in the extent of downstream signaling, as evidenced by stronger IFN-β-induced STAT1/2 phosphorylation in CD5ʰⁱ than in CD5ˡᵒ cells (Fig. 2h).

Given the greater self-reactivity of CD5ʰⁱ cells (Nur77ʰⁱ; Supplementary Fig. 2h)[18], we next examined whether self-TCR signaling indeed affects the sensitivity of T cells to type I IFN. The following three observations support this phenomenon. First, IFN-β-induced Ly6C expression levels were increased significantly upon treatment with soluble anti-CD3 (Supplementary Fig. 2i). Second, Ly6C⁺ cells were poorly synthesized in $Tap1^{-/-}$ hosts that were adoptively transferred with Ly6C⁻ CD8⁺ donor cells (Fig. 2i). Third, most importantly, when Ly6C⁻ B6, P14, and OT-I CD8⁺ T cells were co-transferred to B6 hosts, Ly6C⁺ cell

generation was exactly proportional to the intrinsic self-reactivity of the donor cells (i.e., B6 < P14 < OT-I) (Fig. 2j, left). However, when co-transferred to P14 hosts, P14 donors showed ~2–3 times less Ly6C$^+$ cell generation (presumably reflecting reduced self-interaction due to strong intra-clonal TCR competition for the same self-ligands between the donor and host P14 cells) than that in B6 hosts, whereas OT-I and B6 donors either remained unchanged or even increased (Fig. 2j). In close accord with these findings, OT-I donor cells adoptively transferred either to $Tap1^{-/-}$ hosts or to OT-I hosts showed significantly reduced Ly6C$^+$ cell generation and IFN-β-induced STAT1 phosphorylation (Supplementary Fig. 2j, k). These results strongly suggest that the self-TCR signal is crucial for promoting type I IFN sensitivity and serves as a positive regulator of type I IFN-induced Ly6C$^+$ cell generation.

Since type I IFN is a pro-inflammatory cytokine primarily produced during pathogen infection, the type of stimuli that induced type I IFN production even during the steady state remained undetermined. Ly6C$^+$ naive CD8$^+$ T cells seen in conventional specific pathogen-free (SPF) mice were similarly observed in mice raised in germ-free (GF) and antigen-free (AF) conditions (Supplementary Fig. 2l), suggesting a role of self, but not a foreign, component. Collectively, we propose that naive CD8$^+$ T cell pools continuously receive a tonic signal from type I IFN and self-ligands. This mechanism can be attributed to CD5$^{hi}$ cells, which are likely more sensitive to these cues than CD5$^{lo}$ cells and serve as the main producer of the Ly6C$^+$ subset.

**Type I IFN affects the transcriptome of naive CD8$^+$ T cells in a steady state**. To further understand the effect of tonic type I IFN, we purified B6 naive CD8$^+$ CD5$^{lo}$, Ly6C$^-$, and Ly6C$^+$ cells (Supplementary Fig. 3a) and then compared their gene expression profiles by RNA-Seq analysis. A number of genes were either up- or downregulated in CD5$^{hi}$ cells compared to CD5$^{lo}$ cells (Fig. 3a). Approximately 56–75% of these genes were regulated to similar degrees in both Ly6C$^-$ and Ly6C$^+$ subsets, while ~15% were regulated more in Ly6C$^+$ than in Ly6C$^-$ cells, and ~9–30% were regulated more in Ly6C$^-$ than in Ly6C$^+$ cells (Fig. 3b). Similar results were observed in the microarray analysis (Supplementary Fig. 3b, c). In particular, the $Tbx21$, $Eomes$, $Il18rap$, and $Ccl5$ genes, which are known to have higher expression in CD5$^{hi}$ than in CD5$^{lo}$ cells[7,10], were expressed at levels ~2–5 times higher in the Ly6C$^+$ subset than in the Ly6C$^-$ subset (Fig. 3c). A similar increase was observed in the thymic Ly6C$^+$ subset compared to the Ly6C$^-$ subset (from CD5$^{hi}$C-D24$^{lo}$CD4$^-$CD8$^+$ SP thymocytes; Supplementary Fig. 3d). These data suggest that the Ly6C$^+$ subset is predisposed to acquire unique transcriptomes, beginning from the SP stage during thymic development.

Next, we investigated whether the different transcriptomes of the Ly6C$^-$ and Ly6C$^+$ subsets result from differential sensitivity to type I IFN. As determined by gene set enrichment analysis (GSEA), Ly6C$^+$ cells demonstrated higher enrichment scores for genes associated with the regulation of cytokine responses, especially type I IFN responses, than Ly6C$^-$ cells, with CD5$^{lo}$ cells showing the lowest enrichment scores (Fig. 3d, e and Supplementary Fig. 3e–g). Thus, some, but not all, genes that are differentially regulated in Ly6C$^+$ (relative to Ly6C$^-$) cells seem to reflect relatively high responsiveness to tonic type I IFN signaling. To test this hypothesis, naive CD8$^+$ T cells were cultured for 1 day with IFN-β, and RNA-Seq was performed. Subsequently, the resulting data were reanalyzed in comparison with the RNA-Seq data shown in Fig. 3a. Approximately 30% of genes up- or downregulated in CD5$^{hi}$ cells (including Ly6C$^-$ and Ly6C$^+$

subsets) compared to those in CD5$^{lo}$ cells overlapped with genes regulated by IFN-β treatment (Fig. 3f). The overlapping genes indeed showed high enrichment scores for those regulated in a similar manner by IFN-β (Fig. 3g).

To obtain insights into how the above transcriptomic data are related to the functional properties of CD5$^{lo}$, Ly6C$^-$, and Ly6C$^+$ cells and their responsiveness to type I IFN, we utilized public GO and IMMGEN RNA-Seq databases linking various T cell functional responses[10,19]. Ly6C$^+$ cells showed higher gene enrichment scores with several GO- and IMMGEN-defined functional responses than CD5$^{lo}$ cells and even Ly6C$^-$ cells (Fig. 3h). Of these, clusters I and III (related to effector response, cell division, and proliferation) were significantly influenced by type I IFN (Table 1 and Supplementary Table 1; Eq. 1 and Eq. 2). Together, these data strengthen the perspective that naive CD8$^+$ T cells are variably responsive to tonic type I IFN depending on their intrinsic self-reactivity, shaping not only surface phenotypes but also transcriptomes, perhaps resulting in differential functional properties.

**Ly6C$^+$ cells exhibit enhanced functional properties in a type I IFN-dependent manner**. Given the different effects of type I IFN on transcriptomes, we investigated whether CD5$^{lo}$, Ly6C$^-$, and Ly6C$^+$ cells indeed differ in their functional properties. For this, we examined the innate function of naive CD8$^+$ T cells[20]. B6 splenocytes were cultured for 12–24 h with IL-12 and IL-18 (±IL-2), and IFN-γ production was measured in CD5$^{lo}$, Ly6C$^-$, and Ly6C$^+$ cells (Fig. 4a, top). Ly6C$^+$ cells produced higher amounts of IFN-γ than CD5$^{lo}$ cells and even Ly6C$^-$ cells (Fig. 4a), although the level of such response of Ly6C$^+$ cells was far much lower than that of CD44$^{hi}$ MP cells (Supplementary Fig. 4a). Similar results were observed in P14 CD8$^+$ counterparts (Supplementary Fig. 4b). However, the thymic Ly6C$^+$ subset failed to produce IFN-γ in response to IL-12/IL-18 (Supplementary Fig. 4c), suggesting post-thymic programming with respect to this attribute.

Next, we examined whether the enhanced response of Ly6C$^+$ cells depends on type I IFN, WT and $Ifnar^{-/-}$ splenocytes were cultured with IL-12/IL-18 and analyzed for IFN-γ production (Fig. 4b, top). It is important to note that CD5$^{lo}$ cells were compared with CD5$^{hi}$ cells, as Ly6C$^+$ naive CD8$^+$ cells are absent in $Ifnar^{-/-}$ mice. WT and $Ifnar^{-/-}$ CD5$^{lo}$ cells showed no difference in IFN-γ production (Fig. 4b). However, in CD5$^{hi}$ cells, IFN-γ production was significantly lower in $Ifnar^{-/-}$ cells than in WT cells (~26–50% reduction; Fig. 4b).

The reduced IFN-γ-producing ability in $Ifnar^{-/-}$ CD5$^{hi}$ cells raises the question of how long these cells need to be exposed to type I IFN to be fully functional. Since a fraction of Ly6C$^-$ cells were converted to Ly6C$^+$ cells within a week in the periphery (Fig. 2d), we examined whether type I IFN-driven newly induced (in) Ly6C$^+$ cells (inLy6C$^+$) also acquire enhanced functional capacity. For this, purified Ly6C$^-$ naive CD8$^+$ cells were transferred to B6 hosts, and 7 days later, splenocytes were treated with IL-12/IL-18 and analyzed for IFN-γ production (Fig. 4c, top). Type I IFN-induced donor-derived Ly6C$^+$ cells (inLy6C$^+$) demonstrated IFN-γ production comparable (or moderately lower) to that of pre-existing host-derived Ly6C$^+$ cells (host Ly6C$^+$), both of which demonstrated higher IFN-γ production than that by CD5$^{lo}$ and Ly6C$^-$ donors (Fig. 4c).

To further confirm the above findings, B6 splenocytes derived from mice that were deprived of type I IFN signaling in vivo by treatment with anti-IFNAR were cultured with IL-12/IL-18 and analyzed for IFN-γ production (Fig. 4d, top). Anti-IFNAR-treated Ly6C$^+$ (but not CD5$^{lo}$ or Ly6C$^-$) cells showed an ~18–47% reduction in IFN-γ production compared to that in control

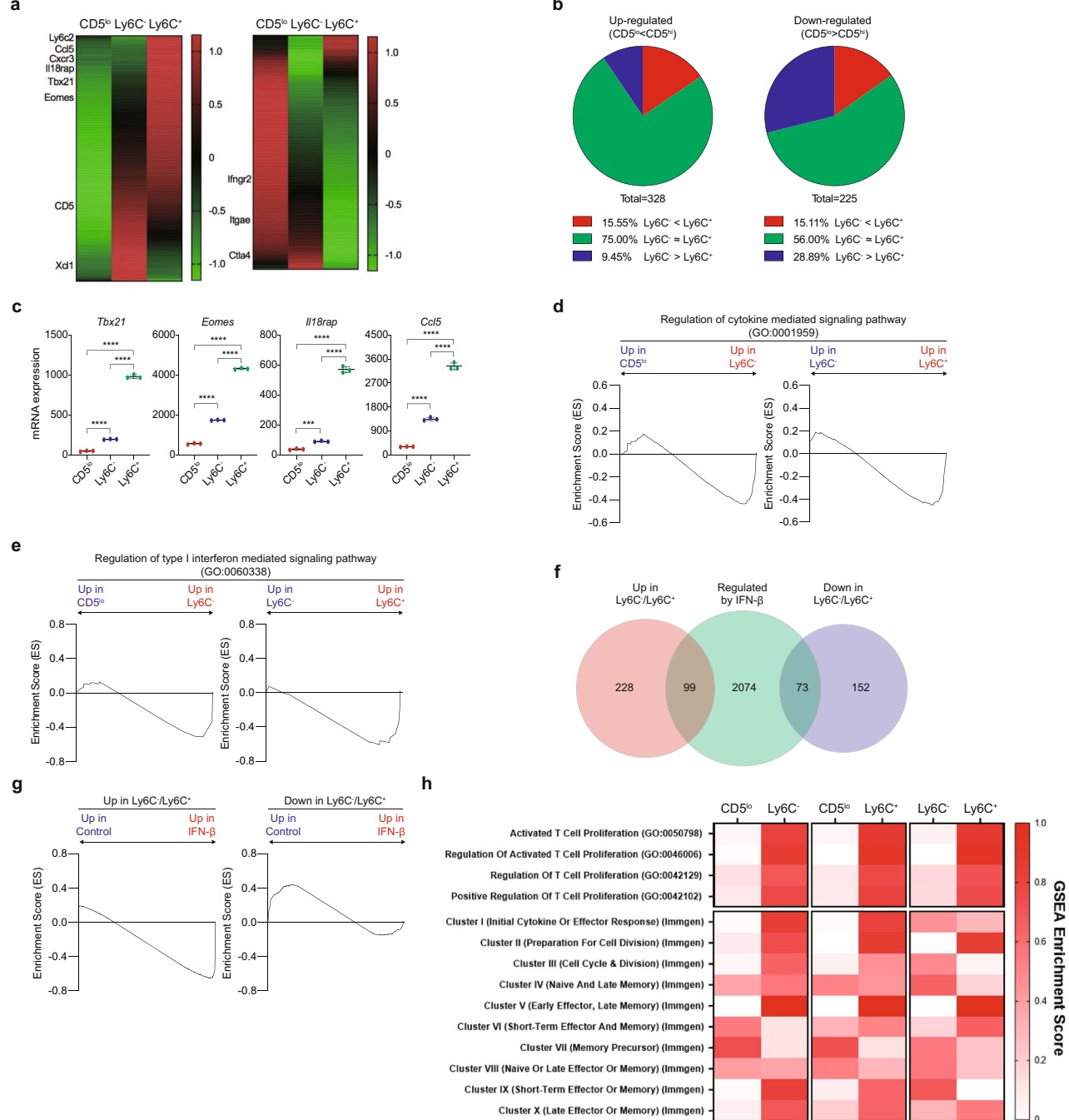

**Fig. 3 Type I IFN affects transcriptomes of naive CD8+ T cells in a steady-state. a** Heat map of genes that are either up-regulated ($n = 328$ genes) (left panel) or downregulated ($n = 225$ genes) (right panel) in FACS-purified B6 naive CD5hi (Ly6C− and Ly6C+) CD8+ T cells compared to CD5lo cells. The color scale is based on Z-score scaling from -1.2 (Green) to 1.2 (Red). **b** Pie charts illustrating genes that are regulated similarly or differently in Ly6C− and Ly6C+ cells in up- (left panel) or downregulated (right panel) genes. **c** RT-PCR results of *Tbx21, Eomes, Il18rap,* and *Ccl5* genes in CD5lo, Ly6C− and Ly6C+ cells. Y-axis represent relative mRNA expression ($n = 3$ mice for each group). Statistical significance was confirmed by two-tailed unpaired Student's t-test. Results are shown as mean ± SD. Data representative of two independent experiments. **d, e** GSEA results between CD5lo and Ly6C−, or Ly6C− and Ly6C+ cells for pathway: Regulation of cytokine mediated signaling pathway (GO:0001959) (**d**) and Regulation of type I interferon mediated signaling pathway (GO:0060338) (**e**). **f** Venn diagram between up- or downregulated genes in CD5hi cells and IFN-β regulated genes. **g** GSEA results between cells cultured with or without IFN-β for up- (right panel) or downregulated (left panel) genes. **h** GSEA were performed between CD5lo and Ly6C−, CD5lo, and Ly6C+, or Ly6C− and Ly6C+ for indicated pathways. GSEA Enrichment Score (ES) represent absolute value of either minimum or maximum running enrichment score. The color scale is based on ES, scaling from 0 (White) to 1.0 (Red). Source data are provided as a Source data file.

Ab-treated Ly6C+ cells (Fig. 4d). Furthermore, in agreement with the finding that a self-TCR signal is a positive regulator for tonic type I IFN response, naive CD8+ T cells transferred to *Tap1*−/− hosts showed significant reduction of IL-12/IL-18-induced IFN-γ production compared to cells transferred to B6 hosts

(Supplementary Fig. 4d). Together, these data suggest that the enhanced functional fitness of Ly6C+ cells is self-driven and acquired post-thymically, at least in part, via a mechanism dependent on prolonged continuous exposure to tonic type I IFN in a steady state.

**Table 1 Influence of IFN-β on core enrichment genes of Immgen clusters between CD5$^{lo}$ and CD5$^{hi}$ naive CD8$^+$ T cells.**

| Immgen Cluster | # of genes in cluster | # of core enrichment genes | Expected # of IFNβ regulated core enrichment genes | Actual # of IFNβ regulated core enrichment genes | P-value |
|---|---|---|---|---|---|
| Cluster I (Initial cytokine or effector response) | 16 | 6 | 0.61 | 6 | 2.26E−06 |
| Cluster II (Preparation for cell division) | 149 | 80 | 8.17 | 3 | 6.17E−02 |
| Cluster III (Cell cycle & division) | 155 | 87 | 8.89 | 35 | 3.97E−13 |
| Cluster IV (Naive and late memory) | 5 | 2 | 0.20 | 1 | 3.88E−01 |
| Cluster V (Early effector, late memory) | 16 | 13 | 1.33 | 0 | 4.93E−01 |
| Cluster VI (Short-term effector and memory) | 34 | 6 | 0.61 | 2 | 2.37E−01 |
| Cluster VII (Memory precursor) | 29 | 11 | 1.12 | 3 | 1.88E−01 |
| Cluster VIII (Naive or late effector or memory) | 76 | 21 | 2.15 | 3 | 7.29E−01 |
| Cluster IX (Short-term effector or memory) | 7 | 3 | 0.31 | 1 | 5.52E−01 |
| Cluster X (Late effector or memory) | 16 | 5 | 0.51 | 1 | 8.33E−01 |

Expected number of IFN-β regulated core enrichment genes and p-values are calculated using Eq. 1 and Eq. 2, respectively.

**Ly6C$^+$ cells induce more antigen-specific expansion than Ly6C$^-$ cells in response to LCMV infection.** Given the markedly different functional capacities, we tested actual immune responses to pathogen infection. For this, B6 naive CD5$^{lo}$, Ly6C$^-$, and Ly6C$^+$ CD8$^+$ T cells were co-transferred to B6 hosts (group 1, CD5$^{lo}$ + Ly6C$^-$; group 2, CD5$^{lo}$ + Ly6C$^+$; and group 3, Ly6C$^-$ + Ly6C$^+$ at a 1:1 ratio), followed by lymphocytic choriomeningitis virus (LCMV) infection and analysis of donor expansion on day 7 post-infection (7 dpi) (Fig. 5a, top). Overall, CD8$^+$ donor cells, involving both Ly6C$^-$ and Ly6C$^+$ cells, demonstrated more expansion than CD5$^{lo}$ cells that were co-transferred (Fig. 5a, upper groups 1 and 2), which was consistent with the results of a prior study performed using total unfractionated CD5$^{hi}$ cells[7]. Surprisingly, when comparing the responses between fractionated CD5$^{hi}$ Ly6C$^-$ and Ly6C$^+$ cells, Ly6C$^+$ cells showed much more expansion than Ly6C$^-$ cells (Fig. 5a, upper group 3). Similar data were observed for LCMV GP33- and NP396-specific CD8$^+$ donor cells (Fig. 5a, lower groups 1–3 and Supplementary Fig. 5a).

The higher expansion rate of Ly6C$^+$ cells compared with Ly6C$^-$ and CD5$^{lo}$ cells was not due to different TCR avidities for cognate peptides, as evidenced by the near-identical TCR binding to GP33- or NP396-MHC tetramers (Supplementary Fig. 5b). To further rule out subtle differences in TCR specificity and precursor frequency of polyclonal B6 CD8$^+$ donors, we utilized P14 CD8$^+$ cells. P14 (on a *Rag1$^{-/-}$* background) naive CD5$^{lo}$, Ly6C$^-$, and Ly6C$^+$ (either CD183$^-$ or CD183$^+$) subsets were co-transferred to B6 hosts (group 1, CD5$^{lo}$ + Ly6C$^-$; group 2, CD5$^{lo}$ + Ly6C$^+$CD183$^-$; and group 3, CD5$^{lo}$ + Ly6C$^+$CD183$^+$ at a 1:1 ratio), followed by LCMV infection and analysis at 8 dpi (Fig. 5b, top). Despite the same TCR specificity and input cell numbers, P14 CD5$^{lo}$ cells recovered to a significantly lesser extent than their P14 CD5$^{hi}$ counterparts (Fig. 5b, group 1/2/3). However, not all CD5$^{hi}$ subsets demonstrated such an increase (Fig. 5b, c, group 1, 2, and 3); compared to CD5$^{lo}$ cells, Ly6C$^+$ (both CD183$^-$ and CD183$^+$) cells showed a significant increase, whereas the Ly6C$^-$ subset did not.

To avoid unnecessary competition between donor subsets co-transferred to the same hosts, P14 naive CD5$^{lo}$, Ly6C$^-$, and

Ly6C$^+$ cells were transferred separately to B6 hosts, followed by LCMV infection and analysis at 7 dpi (Fig. 5d, top). Again, Ly6C$^+$ cells showed greater expansion than CD5$^{lo}$ cells and even Ly6C$^-$ cells, whereas Ly6C$^-$ cells did not (Fig. 5d and Supplementary Fig. 5c). The increased expansion of Ly6C$^+$ cells was associated with enhanced cell division, as evidenced by increased BrdU uptake after 2 h of BrdU pulsing in vivo at 6 dpi (Fig. 5e), increased Ki-67 expression (Supplementary Fig. 5d), and increased Cell Trace Violet (CTV) dye dilution (Supplementary Fig. 5e). Collectively, these data suggest that despite identical TCR and precursor frequencies, the Ly6C$^+$ subset differs from the Ly6C$^-$ subset with respect to the induction of antigen-specific expansion, contributing as a main responder among CD5$^{hi}$ cells overall.

**Ly6C$^+$ cells newly induced by type I IFN exhibit enhanced antigen-specific expansion upon LCMV infection.** The increased response of Ly6C$^+$ cells relative to Ly6C$^-$ cells is intriguing since they are the same P14-derived CD5$^{hi}$ cells. Again, neither subset exhibited any difference in GP33 peptide/MHC tetramer binding (Supplementary Fig. 5f, g). Thus, the greater expansion of Ly6C$^+$ cells compared to Ly6C$^-$ cells may not be fully explained by the differences in TCR avidity for cognate foreign (and self) peptides.

Alternatively, we tested the role of type I IFN owing to the superior reactivity of the Ly6C$^+$ subset to this cytokine and examined whether type I IFN-driven newly induced Ly6C$^+$ cells acquire a better expansion capacity in vivo. Ly6C$^-$ P14 cells were transferred to and parked in B6 hosts for 7 and 21 days to generate type I IFN-induced Ly6C$^+$ cells (d7- and d21-inLy6C$^+$, respectively; Fig. 5f, top). The donor-derived CD5$^{lo}$, Ly6C$^-$, and d7- and d21-inLy6C$^+$ cells, and as a control, pre-existing Ly6C$^+$ cells freshly isolated from P14 mice (freshLy6C$^+$) were purified, transferred to B6 hosts, infected with LCMV, and analyzed for donor expansion at 7 dpi (Fig. 5f, top). While CD5$^{lo}$ and Ly6C$^-$ donors showed similar responses, inLy6C$^+$ cells showed disparate responses depending on the in vivo exposure time (Fig. 5f,

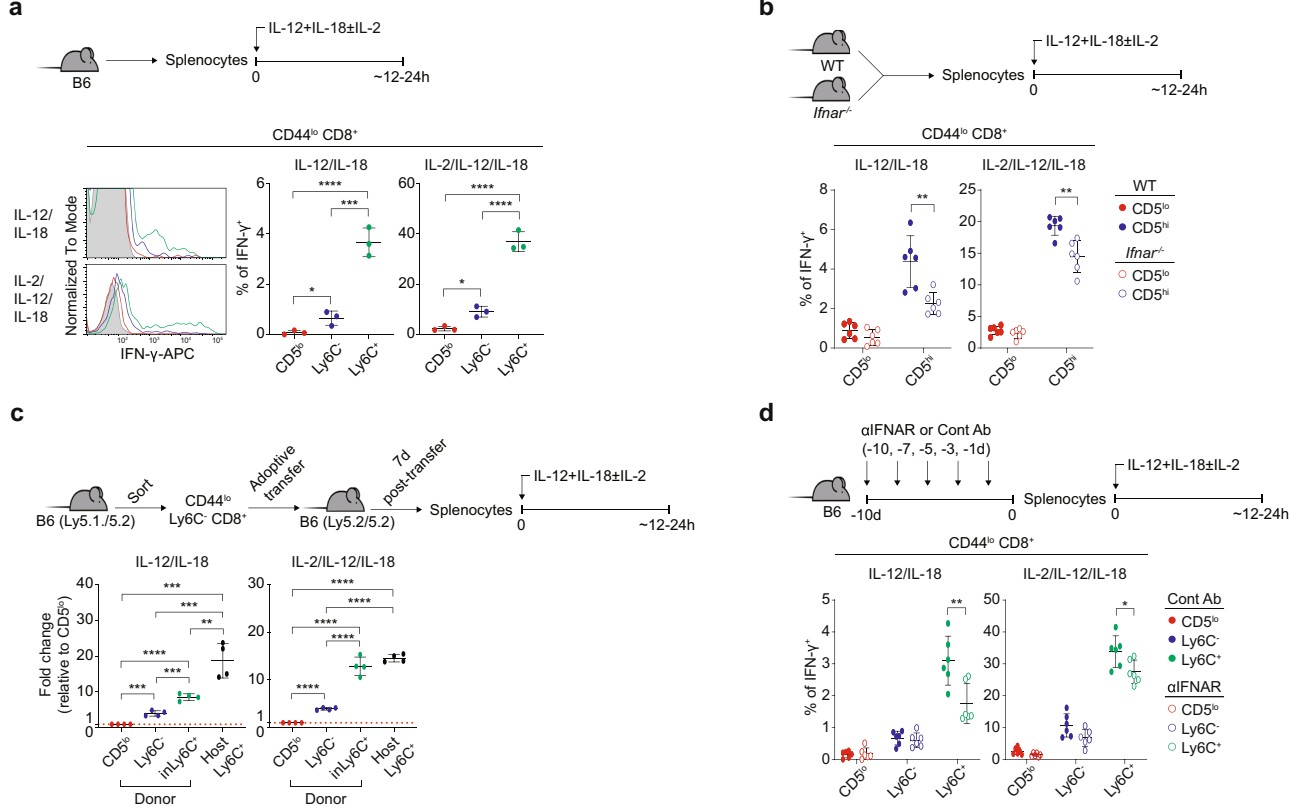

**Fig. 4 Type I IFN affects functional properties of Ly6C+ naive CD8+ T cells. a** IFN-γ production of CD5lo, Ly6C−, and Ly6C+ B6 naive CD8+ T cells (n = 3 mice for each group). **b** IFN-γ production of CD5lo and CD5hi naive CD8+ T cells from WT or Ifnar−/− mice (n = 6 mice for each group). **c** Relative IFN-γ production of CD5lo, Ly6C−, induced Ly6C+ (inLy6C+) donor cells and host Ly6C+ cells (n = 4 mice for each group). **d** IFN-γ production of CD5lo, Ly6C−, and Ly6C+ B6 naive CD8+ T cells pretreated with Control Ab or anti-IFNAR1 Ab in vivo (n = 6 mice for each group). Statistical significance was confirmed by two-tailed unpaired Student's t-test. Results are shown as mean ± SD. Data representative of 2–3 independent experiments. Source data are provided as a Source data file.

bottom). Thus, d21-inLy6C+ donor cells, but not d7-inLy6C+ cells, showed significantly increased antigen-specific expansion compared to that of the Ly6C− donor, which was comparable to that of host-derived pre-existing freshLy6C+ cells (Fig. 5f, bottom), indicating that the higher response of Ly6C+ subset is not directly correlated to the Ly6C expression per se but rather reflects the duration of its in vivo residency. Together, these data suggest that the enhanced expansion capacity of Ly6C+ cells is acquired post-thymically in a time-dependent manner, presumably requiring a prolonged exposure of a tonic type I IFN signal.

**CD5lo, Ly6C−, and Ly6C+ cells are subjected to unique differentiation fates in response to LCMV infection.** Based on the different expansion responses, we examined the fates of effector cells, such as short-lived effector cells (SLECs) vs. memory precursor effector cells (MPECs), at the peak of LCMV-specific immune responses[21]. P14 naive CD5lo, Ly6C−, and Ly6C+ subsets were transferred to B6 hosts, followed by infection with LCMV and analysis of donor differentiation at 7 dpi (Fig. 6a). The percentage of SLECs (CD127loKLRG1hi) was the lowest in CD5lo cells, intermediate in Ly6C− cells, and highest in Ly6C+ cells (Fig. 6b). Conversely, the percentage of MPECs (CD127hiKLRG1lo) was the highest in CD5lo cells, intermediate in Ly6C− cells, and lowest in Ly6C+ cells (Fig. 6b). Similar results were observed based on CD27 and CX3CR1 expression[22] (Fig. 6c); the percentages of CD27−CX3CR1hi (SLEC skewing) and CD27+CX3CR1lo (MPEC skewing) were the highest in Ly6C+ cells and CD5lo cells,

respectively. This phenomenon was also confirmed in experiments with B6 CD8+ subsets as a donor, with the highest and lowest proportion of KLRG1hi cells in Ly6C+ cells and CD5lo cells, respectively, at 7 dpi (Supplementary Fig. 6a).

To obtain more evidence supporting such disparate fates, RNA-Seq was performed on P14 donor subsets (7 dpi) and compared with public RNA-Seq datasets analyzed for SLECs and MPECs[23,24] to proceed with GSEA (Fig. 6d, e). Ly6C+ cells showed more enrichment of SLEC signature genes than CD5lo and Ly6C− cells (Fig. 6d). In contrast, CD5lo cells showed more enrichment of MPEC signature genes than Ly6C− and Ly6C+ cells (Fig. 6e). These data strongly suggest that despite exhibiting the same P14 TCR response to the same cognate antigen, Ly6C+ and CD5lo subsets undergo distinctly different differentiation programs, i.e., SLECs vs. MPECs, respectively.

To further investigate memory cell generation, P14 naive CD5lo, Ly6C−, and Ly6C+ cells were transferred to B6 hosts, followed by LCMV infection and analysis at 120 dpi (Fig. 6f, top). In line with enhanced MPEC skewing, CD5lo cells showed a higher percentage and number than Ly6C− and Ly6C+ cells (Fig. 6f), which was in sharp contrast to the peak response observed at 7 dpi (Fig. 5d). Moreover, CD5lo and Ly6C+ donor cells at 120 dpi showed higher CD44hiCD62Lhi central memory (TCM) and CD44hiCD62Llo effector memory (TEM) phenotype, respectively (Fig. 6g). The enhanced ability of CD5lo cells to differentiate into TCM over TEM was associated with their higher self-renewal capacity, as evidenced by the increased BrdU uptake of P14 CD5lo donor-derived memory cells at 35 dpi compared to

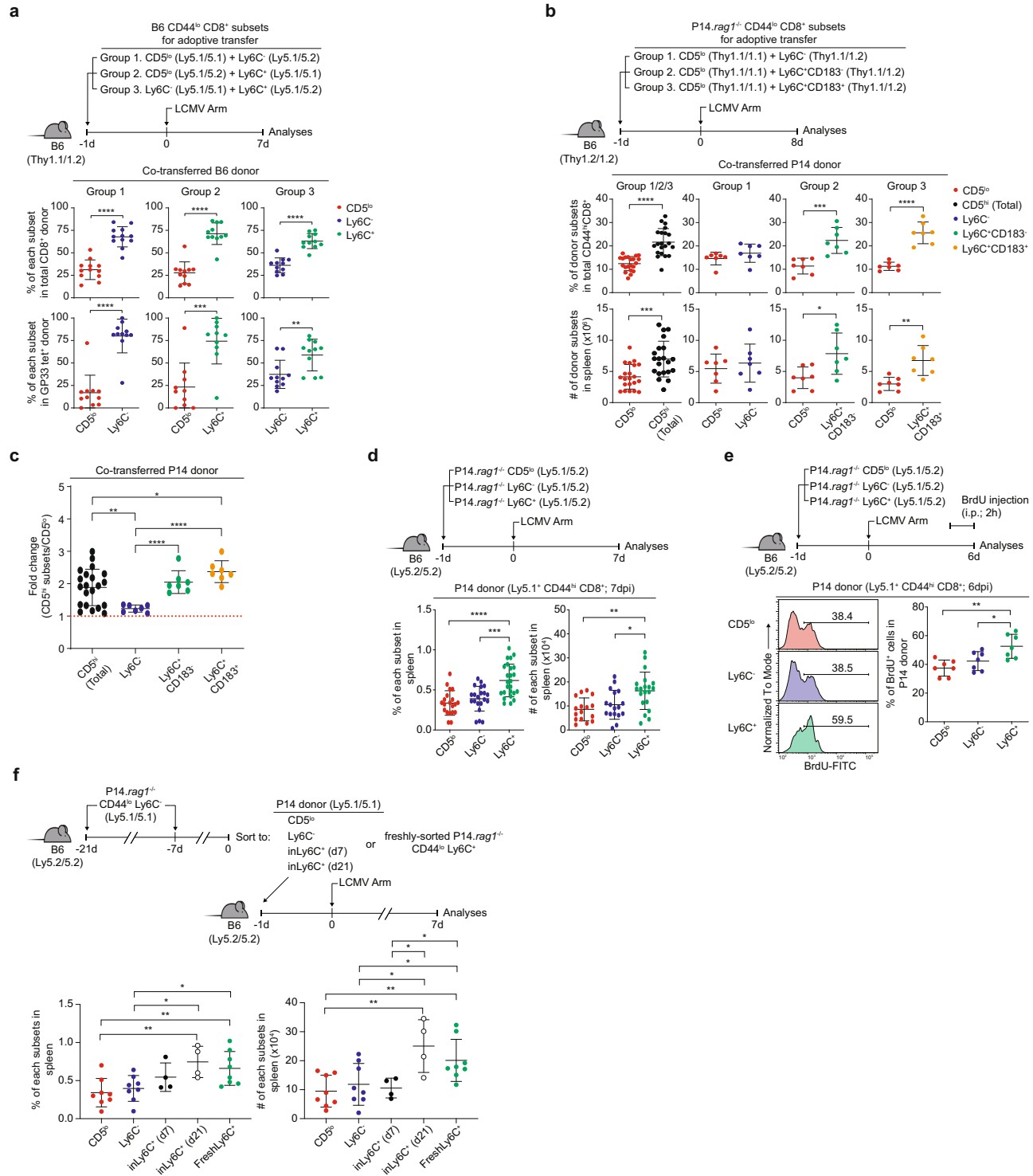

**Fig. 5 Ly6C⁺ cells exhibit greater expansion capacity upon LCMV infection. a** Proportion of each donor subsets within CD44ʰⁱ CD8⁺ (upper) or CD44ʰⁱ CD8⁺ GP33- tetramer⁺ B6 donor cells (lower) (n = 12 mice for Group1, n = 11 mice for Group2 and Group3). **b** Percentage and donor recovery of co-transferred P14 subsets (n = 21 mice for Group 1/2/3, n = 7 mice for Group1, Group2, and Group3). **c** Donor recovery of P14 CD5ʰⁱ subsets relative to P14 CD5ˡᵒ cells (n = 21 mice for CD5ʰⁱ (Total), n = 7 mice for Ly6C⁻, Ly6C⁺ CD183⁻, and Ly6C⁺ CD183⁺). **d** Percentage and donor recovery of individually transferred P14 subsets (n = 20 mice for CD5ˡᵒ, n = 21 mice for Ly6C⁻, n = 25 mice for Ly6C⁺). **e** Percentage of P14 CD44ʰⁱCD8⁺BrdU⁺ donor cells (n = 7 mice for each group). **f** Percentage and donor recovery of P14 subsets previously parked in B6 hosts (n = 8 mice for CD5ˡᵒ, n = 8 mice for Ly6C⁻, n = 4 mice for inLy6C⁺ (d7) and inLy6C⁺ (d21), n = 8 mice for FreshLy6C⁺). Statistical significance was confirmed by two-tailed unpaired Student's t-test. Results are shown as mean ± SD. Data representative of 2–3 independent experiments. Source data are provided as a Source data file.

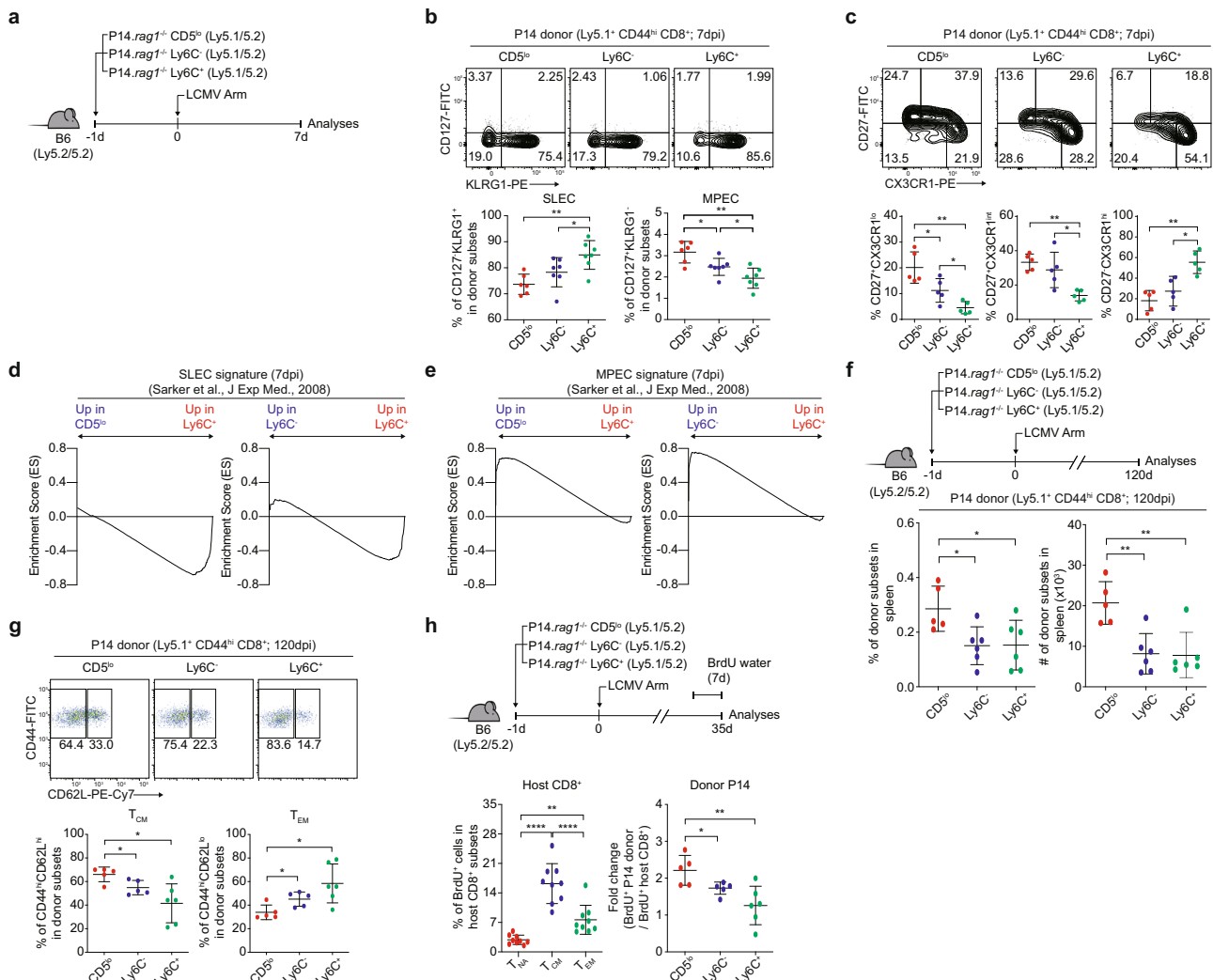

**Fig. 6 CD5^{lo}, Ly6C^−, and Ly6C^+ cells exhibit distinct effector fates upon LCMV infection. a** Experimental scheme of (**b–f**). **b** Percentage of CD127^−KLRG1^+ and CD127^+KLRG1^− P14 CD44^{hi} donor cells (n = 6 mice for CD5^{lo}, n = 7 mice for Ly6C^− and Ly6C^+). **c** Percentage of CD27^+CX3CR1^{lo}, CD27^+CX3CR1^{int}, and CD27^−CX3CR1^{hi} P14 CD44^{hi} donor cells (n = 5 mice for each group). **d**, **e** GSEA profiles of SLEC (**d**) and MPEC (**e**) signature genes between P14 CD5^{lo} and P14 Ly6C^+ donor cells or P14 Ly6C^− and P14 Ly6C^+ donor cells. **f** Percentage and donor recovery of P14 CD44^{hi} donor cells at 120 dpi (n = 5 mice for CD5^{lo}, n = 6 mice for Ly6C^− and Ly6C^+). **g** Percentage of CD44^{hi} CD62L^{hi} and CD44^{hi} CD62L^{lo} P14 donor cells at 120 dpi (n = 5 mice for CD5^{lo} and Ly6C^−, n = 6 mice for Ly6C^+). **h** Percentage of P14 CD44^{hi} BrdU^+ donor cells at 35 dpi (n = 9 mice for T_{NA}, T_{CM}, and T_{EM}, n = 5 mice for CD5^{lo}, n = 6 mice for Ly6C^− and Ly6C^+). Data are accumulated values of at least two independent experiments (**b–i**). Statistical significance was confirmed by two-tailed unpaired Student's t-test. Results are shown as mean ± SD. Data representative of 2–3 independent experiments. Source data are provided as a Source data file.

their Ly6C^− and Ly6C^+ counterparts (Fig. 6h and Supplementary Fig. 6b). Collectively, these data suggest that the differentiation fate of individual naive CD8^+ T cells is pre-determined inherently regardless of their antigen specificity rather than being randomly selected during immune responses.

**Tonic type I IFN shapes inherently high SLEC but low MPEC skewing of Ly6C^+ cells.** Given the inherently different fates, we examined other intrinsic properties, such as apoptosis, cytokine production, and metabolism. For this, P14 CD5^{lo}, Ly6C^−, and Ly6C^+ cells were transferred to B6 hosts, followed by LCMV infection and analysis at 7 dpi (Fig. 7a). Upon measurement of active caspase 3, CD5^{lo} cells showed a lower death rate than Ly6C^− and Ly6C^+ cells (Fig. 7b). For cytokine production after peptide restimulation in vitro for 5 h, CD5^{lo} cells demonstrated higher IL-2 production than Ly6C^− and Ly6C^+ cells, although IFN-γ and TNF-α production remained unchanged (Fig. 7c and

Supplementary Fig. 7a). No difference was observed in granzyme B or CD107a expression (Supplementary Fig. 7b). Upon measurement of the oxygen consumption rate (OCR) and extracellular acidification rate (ECAR), CD5^{lo} cells demonstrated a lower ECAR and higher OCR than Ly6C^− and Ly6C^+ cells, with an increased OCR:ECAR ratio in CD5^{lo} cells (Fig. 7d). In line with the reduced glycolytic activity (ECAR), CD5^{lo} cells demonstrated a significantly lower gene set enrichment score for the mTOR signaling pathway than Ly6C^− and Ly6C^+ cells (Supplementary Fig. 7c). Thus, the aforementioned characteristics observed in CD5^{lo} cells closely resembled less differentiated MPECs[25,26].

To investigate whether tonic type I IFN in a steady state contributes to the shaping of different fates, we performed an in vivo blocking experiment with anti-IFNAR treatment. P14 mice were injected with control or an anti-IFNAR Ab, and 10 days later, CD5^{lo} and Ly6C^+ cells were isolated and co-transferred to B6 hosts (at a 1:1 mixture of the control Ab- and

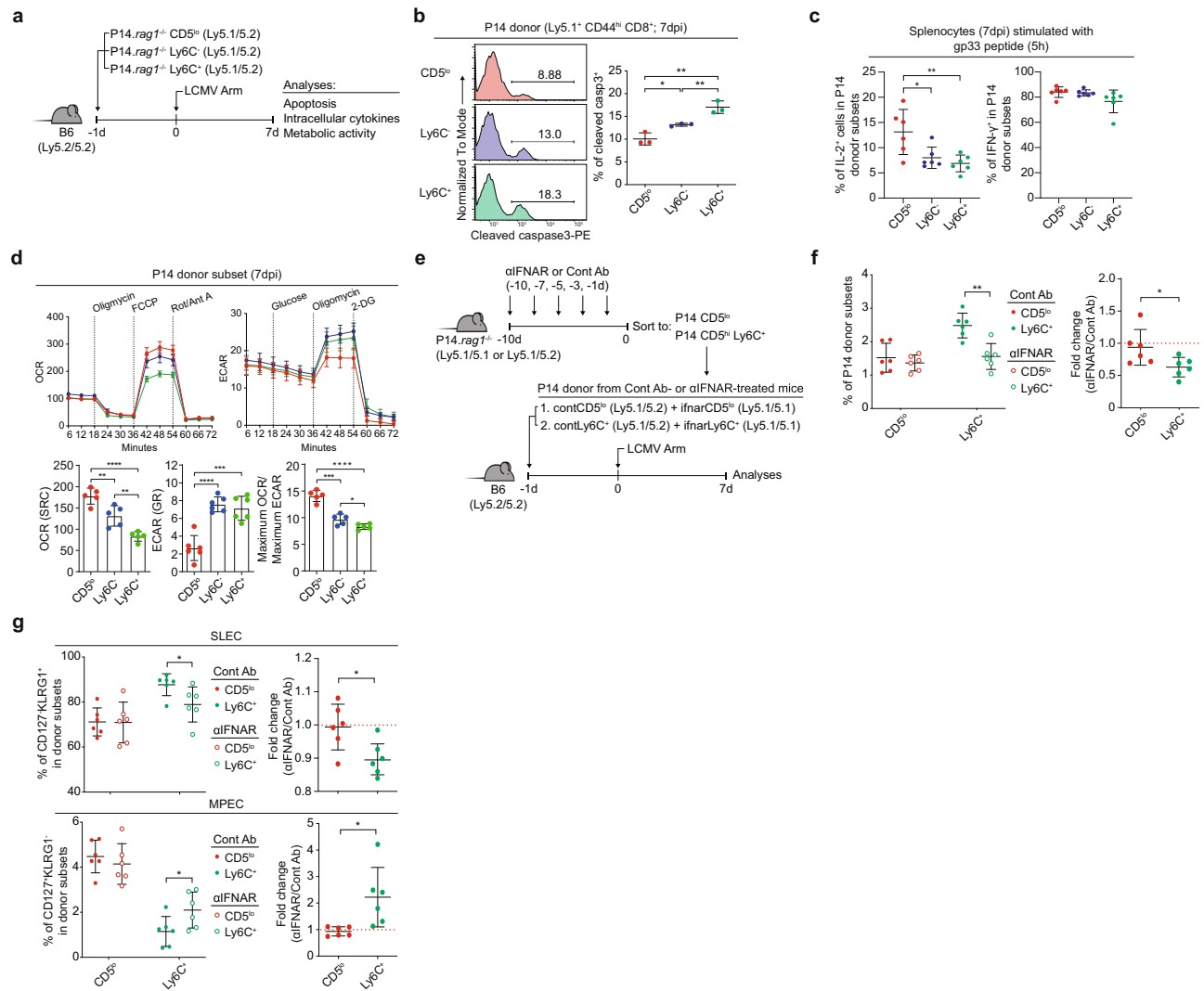

**Fig. 7 Tonic type I IFN shapes the inherent ability of Ly6C⁺ cells to acquire high SLEC but low MPEC skewing upon LCMV infection. a** Experimental scheme of (**b–d**). **b** Percentage of cleaved caspase 3⁺ donor cells ($n = 3$ mice for each group). **c** IL-2 (left) and IFN-γ (right) production of P14 CD44ʰⁱ donor cells ($n = 6$ mice for each group). **d** OCR and ECAR profiles of P14 CD44ʰⁱ donor cells ($n = 5$ mice for OCR (SRC) and Maximum OCR/Maximum ECAR, $n = 6$ mice for ECAR (GR)). **e** Experimental scheme of (**f, g**). **f, g** Percentage of P14 CD44ʰⁱ donor cells (**f**) and Percentage of CD127⁻ KLRG1⁺ or CD127⁺ KLRG1⁻ donor cells (**g**) previously treated with Control Ab or IFNAR Ab in vivo ($n = 6$ mice for each group). Statistical significance was confirmed by two-tailed unpaired Student's *t*-test. Results are shown as mean ± SD. Data representative of 2–3 independent experiments. Source data are provided as a Source data file.

anti-IFNAR-treated CD5ˡᵒ or Ly6C⁺ subset), followed by LCMV infection and analysis at 7 dpi (Fig. 7e). Ly6C⁺, but not CD5ˡᵒ, cells treated with anti-IFNAR showed significantly reduced antigen-specific expansion compared to that of cells treated with the control Ab (Fig. 7f). Likewise, the earlier type I IFN blockade performed for 10 days led to decreased SLEC and conversely increased MPEC skewing of Ly6C⁺ cells, whereas the fate of CD5ˡᵒ cells remained unchanged (Fig. 7g). Similar results were observed with the analysis based on CD27 and CX3CR1 expression (Supplementary Fig. 7d). In sharp contrast to peripheral P14 Ly6C⁺ cells, thymic Ly6C⁺ cells (relative to their thymic CD5ˡᵒ and Ly6C⁻ counterparts) showed no difference in effector expansion or differentiation fate upon LCMV infection (Supplementary Fig. 7e). Collectively, these data suggest that continuous tonic type I IFN exposure in the periphery has a profound effect on shaping the unique differentiation property of naive cells, especially Ly6C⁺ CD8⁺ T cells.

## Discussion

The role of intrinsic self-reactivity has been well demonstrated in naive T cells, highlighting an enhanced immune response in CD5ʰⁱ cells compared to CD5ˡᵒ cells[6–9]. Although coincidentally high TCR affinity for cognate antigen or high intrinsic TCR signaling capacity may explain the enhanced response in CD5ʰⁱ cells, the underlying mechanisms are not fully understood. In this study, we characterized the phenotypic heterogeneity of naive CD8⁺ T cells based on the relative differences in their intrinsic self-reactivity and separated them into CD5ˡᵒLy6C⁻, CD5ʰⁱLy6C⁻, and CD5ʰⁱLy6C⁺ subsets. These subsets exhibited different responses under various in vitro and in vivo conditions. Importantly, the distinct CD5ʰⁱ subsets were not equivalent and were inherently different in their functional fitness in a type I IFN-dependent manner, with Ly6C⁺ cells more prominently demonstrating IFN-γ production, antigen-specific proliferation, and effector differentiation than Ly6C⁻ cells. Therefore, we

conclude that the self-reactivity variably imprints the unique immune property of naive CD8[+] T cells by co-opting tonic type I IFN signaling. This post-thymic programming process is also necessary for eliciting diverse immune responses with CD8[+] T cell pools beyond their TCR specificity.

Peripheral naive CD8[+] T cells are in constant contact with self-ligands throughout their life, which is essential for their survival[27]. In addition, the relative strength of such contact is variable among individual naive T cells and is correlated with their antigen-specific expansion[7,8]. This relationship between self-reactivity and immunity raises an important question of when and how the self-TCR signal shapes T cell function and the subsequent immune response. This phenomenon may come into effect either thymically or post-thymically or perhaps depend on certain physiological cues such as cytokines. In this regard, the physiological role of type I IFN observed in this study was rather unexpected, as this pro-inflammatory cytokine is produced because of innate immune responses to pathogen infection[28,29]. Our study shows that even in the steady state, naive CD8[+] T cells are continuously exposed to type I IFN, which is crucial for the generation of the Ly6C[+] subset from CD5[hi] cells.

Given the higher sensitivity of CD5[hi] cells to type I IFN, which is presumably driven via a process involving a self-TCR signal, such as modulation of membrane microdomains (e.g., lipid rafts)[6] and of intrinsic signaling pathways[9], these cells are more likely to be affected than CD5[lo] cells, thereby resulting in a more complex phenotype including the expression of Ly6C, CD183, and CD103. Among these heterogeneous subsets, Ly6C[+] cells are the most prominent responders to tonic type I IFN and thus express the highest amounts of type I IFN-regulated gene transcripts, especially genes associated with cell division/proliferation. Therefore, a signal from tonic type I IFN is clearly a key player for inducing heterogeneous phenotypes and even transcriptomes preferentially on CD5[hi] cells. As for type I IFN, a similar phenomenon may occur in response to other physiological cytokines, such as IL-7 and IL-15, as they are crucial for T cell homeostasis[30,31]. Thus, future studies will need to address the roles of such additional cues in shaping T cell functional properties.

The enhanced function of Ly6C[+] cells is further emphasized in response to LCMV infection, resulting in higher antigen-specific expansion and SLEC (but less MPEC) differentiation. This phenomenon is independent of TCR specificity but dependent on type I IFN. Thus, the following questions arise: when do Ly6C[+] cells gain this property and how long do these cells need to be exposed to type I IFN to be fully programmed? Our data demonstrated that enhanced expansion and SLEC skewing are prominent in only peripheral Ly6C[+] cells and not thymic Ly6C[+] or Ly6C[+] cells newly induced for a short period from Ly6C[−] cells. Hence, the unique immune responses of Ly6C[+] cells do not depend on Ly6C expression per se but rely on the strong persistent reaction to tonic type I IFN signaling. Since this response to type I IFN is a self-driven process and requires relatively strong self-reactivity, whether the Ly6C[+] cells that are deprived of self-ligands exhibit reduced antigen-specific expansion and SLEC differentiation will be interesting to explore.

The enhanced innate function (i.e., IL-12/IL-18-induced IFN-γ production) and SLEC differentiation in Ly6C[+] cells closely resembles that of virtual memory (VM) cells[32,33]. VM cells phenotypically and functionally differ from naive cells and antigen-experienced memory cells and are induced owing to homeostatic proliferation (HP) by cytokines, such as IL-15 and IL-4, in an antigen-independent fashion[10,34,35]. A recent study showed that VM cells, unlike antigen-experienced true memory cells, express enhanced levels of the transcription factor Eomes and are generated in a type I IFN-dependent manner[36]. Hence, we speculate that Ly6C[+] cells may serve as naive precursors for

conversion into VM cells. Indeed, this assumption is well supported by our findings demonstrating that Ly6C[+] cells express the highest levels of the *Eomes* gene and are dependent on type I IFN for their generation, innate function, and SLEC differentiation, all of which are the primary characteristics of VM cells[10,20,36]. Nevertheless, it is important to emphasize that, although Ly6C[+] cells exhibit such "VM-like" features, these cells are still phenotypically and functionally naive, based on various parameters examined, including lower levels of CD44 and CD122 expression, anti-CD3/28 (or cytokine)-driven proliferation, and PMA/ionomycin (or IL-12/IL-18)-induced IFN-γ production.

When and how Ly6C[+] cells are converted into VM cells remains in question. A recent study showed that CD8[+] T cells that are generated early in life and persist in adults exhibit unique characteristics with respect to their phenotype and function[20]. One of the key findings in this study demonstrated that nearly all such newborn-derived CD8[+] T cells convert into the CD44[hi]CD62L[hi]CD122[hi]CD49d[−] VM population in adult mice[20]. Moreover, these newborn-derived VM cells demonstrate much higher antigen-specific effector expansion and SLEC differentiation than adult-derived VM cells after bacterial infection. Whether such disparate fates are type I IFN dependent and limited to solely the Ly6C[+] subset of newborn-derived CD8[+] T cells was not addressed in this study. Thus, it is possible that our results with Ly6C[+] cells from adult mice (both B6 and P14) might reflect a fraction of the newborn-originated Ly6C[+] cells that can survive for a long time, albeit extremely rarely, in the periphery of adult mice. Hence, it will be interesting to explore this possibility in the future by conducting an experiment in which Ly6C[+] cells purified from B6 mice reconstituted with adult B6 (or P14) bone-marrow cells are tested.

The enhanced capacity of CD5[lo] cells with respect to MPEC differentiation and memory cell formation is an interesting unexpected result. This unique property of CD5[lo] cells remained unchanged even in the short-term blockade of tonic type I IFN signaling in vivo, which was in sharp contrast to that in Ly6C[+] cells. Although it is unclear exactly how such steady-state type I IFN exposure shapes the fate of naive CD8[+] T cells, the reduced responsiveness of CD5[lo] cells to this nonclassical homeostatic cytokine may promote the MPEC fate simply by counteracting the SLEC fate. In this regard, along with transcriptomic analysis, examining whether the physiological type I IFN signal exerts a differential effect on the epigenetic profiles of each naive CD8[+] T cell subset will be interesting. These additional studies and our data with CD5[lo] cells may provide insight into why the thymic selection process has not evolved to generate only highly self-reactive CD5[hi] cells that demonstrate better functional fitness and are capable of inducing better immune responses. Here, we propose a much broader understanding that CD5[lo] cells must also be selected as a useful player contributing to counterbalancing the SLEC bias of CD5[hi] cells, resulting in increased MPECs and, accordingly, memory cell formation (especially T$_{CM}$) in response to pathogen infection.

One single TCR Tg naive CD8[+] T cell has been shown to differentiate into diverse effector (SLEC and MPEC) and memory (T$_{CM}$, T$_{EM}$, and T$_{RM}$) subsets[37,38]. This strongly suggests that such disparate fates are determined by random stochastic events occurring in the activation and expansion phases during pathogen infection. In the present study, however, our data further extend this prevailing notion by highlighting the fact that the potential of single naive T cells to differentiate into multiple fates is not the same across all naive CD8[+] T cells but varies distinctly from one subset to another. Such unique properties of each subset appear to be regulated independently of the TCR diversity and specificity of individual T cells. Importantly, it is programmed, at

least in part, by continuous exposure to tonic type I IFN. Thus, the previously reported multiple differentiation potential of one single naive CD8$^+$ T cell may reflect the collection of such unique properties of individual T cell clones that are imprinted differently in a steady state.

In summary, naive CD8$^+$ T cells continuously co-opt tonic type I IFN signals with varying sensitivity depending on their relative strength of TCR contacts with self-antigens, which, in turn, shapes their unique functional properties that differ from one subset to another. The consequence of these effects is to broaden and increase the complexity of naive CD8$^+$ T cell pools with respect to immune responses beyond their TCR specificity and diversity. Although the importance of tonic type I IFN, particularly on Ly6C$^+$ cells, is mainly emphasized in this study, it still does not rule out a role of other soluble factors available in a steady-state condition. Hence, future studies are expected to clarify additional layers of functional diversity of individual naive CD8$^+$ T cells with yet identified homeostatic cues.

## Methods

**Mice.** C57BL/6 (B6), B6.SJL (Ly5.1) and B6.PL (Thy1.1) mice were purchased from The Jackson Laboratory. P14, OT-I Thy1.1, $Tap1^{-/-}$, $Rag1^{-/-}$, $Ifnar^{-/-}$, $Ifnar.Ifngr^{-/-}$, and $Stat1^{-/-}$ mice (all on a B6 background) were obtained from POSTECH. P14.$Rag1^{-/-}$ Ly5.1 and P14.$Rag1^{-/-}$ Thy1.1 mice were generated by crossing P14 mice with $Rag1^{-/-}$ and B6.SJL or B6.PL mice. Mice were maintained under specific pathogen-free conditions. Germ-free (GF) and antigen-free (AF) mice are maintained sterilely at POSTECH Biotech Center[39]. Mice were maintained in a 12 h light/dark cycles at 25 °C, within 40% humidity with free access to food and water. Unless described otherwise, all mice were used sex-matched at 8-12 weeks of age, according to protocols approved by the Animal Experimental and Ethic Committee at POSTECH and Chonnam National University (CNU).

**Reagents, antibodies, and flow cytometry.** Recombinant murine IL-2, IL-4, IL-6, IL-7, IL-12, IL-15, IL-21, IL-23, IFN-γ, and TGF-β were purchased from Pepro-Tech. Mouse IFN-β was purchased from PBL Biomedical Laboratories. LCMV GP33 peptide (KAVYNFATM; specific for P14 TCR) was purchased from Bioneer. Cell suspensions of spleen, LN, or thymus were prepared and stained for FACS analysis with the fluorochrome-conjugated antibodies. Detailed information of the antibodies used is depicted in Supplementary Table 2. To stain intracellular Nur77, Ki-67, CD107a, and Granzyme B, splenocytes were stained for cell surface markers, fixed and permeabilized using eBioscience Foxp3/Transcription Factor Staining Buffer Set (eBioscience) and then stained with fluorochrome-conjugated intracellular antibodies. Flow cytometry samples were run using a LSRII or FACSCanto II (BD Biosciences) using FACSDiva software (BD Biosciences) and analyzed by FlowJo software (Tree Star). The gating strategies are depicted in Supplementary Fig. 8.

**In vitro T cell proliferation.** CD44$^{lo}$CD5$^{lo}$, CD44$^{lo}$CD5$^{hi}$, and CD44$^{hi}$ CD8$^+$ T cells were purified and labeled with CellTrace Violet (CTV; ThermoFisher) and plated in anti-CD3 (0.1 or 0.3 μg/ml) and anti-CD28 (2 μg/ml) coated Nunc MaxiSorp Flat-Bottom Plate (Thermo Fisher Scientific) and 96-well cell culture plates (SPL) for TCR and IL-15 stimulation respectively. Cells were harvested after 2 or 4 days and CTV dilution was analyzed by flow cytometry.

**In vitro Ly6C induction.** CD44$^{lo}$Ly6C$^-$ CD8$^+$ T cells were purified from pulled LNs using MoFlo Astrios or MoFlo XDP (Beckman Coulter). Purified T cells were plated in 96-well cell culture plates (SPL) (2 × 10$^5$ cells/well) and subjected to the following cytokines: IFN-β (0.001–10 ng/ml), IL-2 (10 ng/ml), IL-4 (10 ng/ml), IL-6 (10 ng/ml), IL-7 (10 ng/ml), IL-12 (10 ng/ml), IL-15 (10 ng/ml), IL-21 (10 ng/ml), IL-23 (10 ng/ml), TGF-β (10 ng/ml), IFN-γ (10 ng/ml)). In some experiments, soluble anti-CD3 (1 μg/ml) was added to culture. Cells were harvested after 24 hours and analyzed by flow cytometry. To analyze restricted Ly6C induction on CD5$^{hi}$ naive CD8$^+$ T cells, CD5$^{lo}$ and Ly6C$^-$ naive CD8$^+$ T cells were purified and incubated with IFN-β (0.1 ng/ml) for 24 h. Cells were harvested and Ly6C induction was analyzed by flow cytometry and quantitative real-time PCR.

**In vivo Ly6C induction.** CD44$^{lo}$Ly6C$^-$ CD8$^+$ T cells were purified from pulled spleens and LNs of indicated mice. Mixture of purified cells from B6 Thy1.1/1.2 and $Ifnar^{-/-}$ Thy1.2/1.2 mice (5 × 10$^5$ cells from each donor) were transferred intravenously (i.v.) to B6 Ly5.1/5.1 hosts. Spleens and LNs were pulled after 7 days and analyzed by flow cytometry. To analyze influence of self-reactivity on Ly6C induction, purified CD44$^{lo}$Ly6C$^-$ CD8$^+$ T cells from B6 Thy1.1/1.2 mice were transferred to B6 or $Tap1^{-/-}$ hosts (5 × 10$^5$ cells/mouse). In some experiments, mixture of purified CD44$^{lo}$Ly6C$^-$ CD8$^+$ T cells from B6 Ly5.1/5.1, P14 Ly5.1/5.2,

and OT-I Thy1.1/1.1 mice (5 × 10$^5$ cells from each donor) were transferred to B6 or P14 hosts. Spleens and LNs were pulled at indicated time points and analyzed by flow cytometry. To compare the functional responses of type I IFN-driven newly induced Ly6C$^+$ subset, purified CD44$^{lo}$Ly6C$^-$ CD8$^+$ T cells either from B6 or P14 mice (~0.5–1 × 10$^6$ cells from each donor) were transferred to B6 hosts and parked for the indicated time points (7–21 days) to newly generate Ly6C$^+$ cells. Total splenocytes (for B6 donors) or purified P14 donor subsets as indicated were tested either for innate function with IL-12/IL-18 stimulation in vitro (for B6 donors) or for antigen-specific expansion in vivo after adoptive transfer and subsequent infection with LCMV (for P14 donors).

**In vitro stimulation.** To analyze innate function of naive CD8$^+$ T cell subsets, splenocytes or thymocytes from indicated mice were plated in 96-well cell culture plates (1 × 10$^6$ cells/well) and cultured for 12–24 h with IL-12 (10 ng/ml), IL-18 (10 ng/ml), and with or without IL-2 (50 ng/ml). In some experiments, spleens of B6 previously transferred with CD44$^{lo}$Ly6C$^-$ CD8$^+$ Ly5.1/5.2 T cells or spleens of B6 i.p. injected with anti-IFNAR (MAR1-5A3; BioXCell) every other day for 10 days (200–300 μg/once/mice) were used. Cultured cells were harvested and stained for cell surface markers, fixed and permeabilized using BD Cytofix/Cyto-perm buffer (BD Biosciences) and then stained with fluorochrome-conjugated anti-IFN-γ antibody. To analyze cytokine production after peptide restimulation, CD5$^{lo}$, Ly6C$^-$, and Ly6C$^+$ CD8$^+$ T cells were purified from P14.$Rag1^{-/-}$ Ly5.1/5.2 mice and transferred to B6 hosts (10$^4$ cells/mice). Mice were infected with LCMV (2 × 10$^5$ pfu/mice) day after transfer and assessed 7 days later. Splenocytes were prepared from infected mice and plated in 96-well cell culture plate (10$^6$ cells/well). Cells were stimulated with titrated dose of gp33 peptide (10$^{-4}$–10$^{-1}$ μM) for 5 h. GolgiPlug (BD Biosciences) was added after the first hour. Cells were harvested and stained with fluorochrome-conjugated anti-IFN-γ, anti-IL-2, and anti-TNF antibodies as described elsewhere.

**Western blot.** To analyze relative sensitivity to type I IFN, purified CD5$^{lo}$ and Ly6C$^-$ naive CD8$^+$ T cells (3 × 10$^5$ cells) were incubated with IFN-β (0.1 ng/ml) for indicated time points and analyzed with western blot. In brief, cells were harvested and lysed in a lysis buffer (20 mM Tris, pH7.5, 150 mM NaCl, 1 mM EDTA, 1 mM EGTA, 1% Triton X-100, 2.5 mM sodium pyrophosphate, 1 mM β-glycerophosphate, 1 mM Na$_3$VO$_4$, 1 mM PMSF, 1 μg/ml aprotinin and leupeptin). Cell lysates were prepared and resolved by 10% Bis-Tris SDS-PAGE Gel (Invitrogen), then transferred onto nitrocellulose membrane (Invitrogen), blocked with 5% dry non-fat milk in Tris buffered saline (pH 7.4) containing 0.1% Tween-20. The membrane was probed with primary antibodies for overnight in 4 °C on a shaking incubator, followed by respective HRP-conjugated secondary antibodies. Immunoreactivity was detected by SuperSignal West Pico Chemiluminescent Substrate (Thermo Scientific) with LAS-3000 (FujiFilm). Full scan blot is available in the Source Data file.

**Real-time PCR.** CD5$^{lo}$, Ly6C$^-$, and Ly6C$^+$ naive CD8$^+$ T cells were purified from pulled spleens and LNs (or from thymus). RNA was isolated with NucleoZOL (Macherey-Nagel) and cDNA was synthesized with M-MLV reverse transcriptase and oligo dT (TAKARA). Real-time RT-PCR was performed with the TaqMan Gene Expression Master Mix using StepOnePlus Real-Time PCR System (Applied Biosystems) using StepOne Software (Applied Biosystems). Detailed information of the primers used is described in Supplementary Table 3.

**RNA sequencing, microarray, and bioinformatics.** Three different groups were used for RNA sequencing (RNA-seq): (1) Ex vivo B6 CD5$^{lo}$, Ly6C$^-$, and Ly6C$^+$ naive CD8$^+$ T cells; (2) B6 naive CD8$^+$ T cells cultured with or without IFN-β (10 ng/ml) (All samples were treated with 0.5 ng/ml of IL-7 to promote survival); and (3) P14 CD5$^{lo}$, Ly6C$^-$, and Ly6C$^+$ CD8$^+$ T cells extracted at 7 dpi. For RNA seq, at least 1 × 10$^6$ purified cells were used for RNA extraction with a NucleoZOL (Macherey-Nagel). Library preparation was carried out with TruSeq Stranded mRNA Sample Preparation Kit (RS-122-2101~2, Illumina) according to the manufacturer's protocol. RNA sequencing was carried out with NextSeq 500 Sequencing System (SY-415-1001, Illumina) using NextSeq 500 HighOutput kit (150 cycles) Reagent Cartridge (15057931, Illumina). For microarray, at least 1 × 10$^6$ purified cells were used for RNA extraction with a TriZOL (Invitrogen). Library preparation was carried out with Ambion Illumina RNA amplification kit (Ambion) according to the manufacturer's protocol. Microarray was carried out with mouseHT-12 expression v.4 bead array (Illumina) and Amersham fluorolink streptavidin-Cy3 (GE Healthcare Bio-Sciences) using Illumina BeadStudio v3.1.3(Gene Expression Module v3.3.8). Gene Set Enrichment Analysis (GSEA) was performed using GSEA tool (Broad Institute). The following options were selected for analysis: Number of permutations, 1000; Collapse dataset to gene symbols, false; Permutation type, phenotype; Enrichment statistic, weighted; Metric for ranking genes, Diff_Of_Classes; Max size, 500; and Min size, 5. GSEA Enrichment Score represents highest or lowest running Enrichment Score. Genes with fold changes between samples greater than 2 are selected as differentially expressed genes (DEGs). Expression values of DEGs were converted to z-score and used for generating Heatmap by Prism (GraphPad Software). To analyze influence of type I IFN on different Immgen clusters, core enrichment genes determined by

GSEA tool were analyzed. Expected number of accidental inclusions of IFN-β regulated genes in core enrichment genes and *p*-value of IFN-β regulated genes in core enrichment genes were calculated as bellow: Total number of genes, *T*; Number of IFN regulated genes, *B*; Number of core enrichment gene in given Immgen cluster, *C*; Actual number of IFN-β regulated genes in core enrichment genes, *A*; Expected number of accidental inclusions of IFN-β regulated genes, *E*.

$$E = \sum_{i=0}^{C} i \times \frac{{}_BC_i \times {}_{T-B}C_{C-i}}{{}_TC_C} \tag{1}$$

$$p\,\text{value} = 2 \times \sum_{i=A}^{C} \frac{{}_BC_i \times {}_{T-B}C_{C-i}}{{}_TC_C}\ (A>E)$$

$$\text{or } 2 \times \sum_{i=0}^{A} \frac{{}_BC_i \times {}_{T-B}C_{C-i}}{{}_TC_C}\ (A<E) \tag{2}$$

**LCMV infection**. LCMV Armstrong was kindly provided by S.J. Ha (Yonsei University, Seoul, Korea). Naive CD8$^+$ subsets were purified from pulled spleens and LNs (or from thymus) of indicated mice and transferred to B6 hosts ($10^2$–$10^4$ cells/mice). In some experiments, two subsets were co-transferred using different congenic markers. Mice were infected with LCMV ($2 \times 10^5$ pfu/mice) day after transfer and assessed 6–8 days later unless indicated otherwise. Spleens and LNs were harvested and analyzed by flow cytometry. To analyze antigen-specific CD8$^+$ T cells from B6 donors, two-step enrichment method was used. Thy1.1 B6 mice were specifically used as hosts for this experiment. Spleens and LNs were pulled and stained with biotin-conjugated anti-CD4 and anti-B220 antibodies (Invitrogen), followed by anti-biotin MicroBeads (Miltenyi Biotec). Stained cells were purified using LS Column and QuadroMACS (Miltenyi Biotec) as described elsewhere. Flow-through was harvested and stained with biotin-conjugated anti-Thy1.1 antibody (Invitrogen), followed by anti-biotin MicroBeads. Cells were purified and the flow-through, now enriched with donor cells, were harvested and stained with GP33- or NP396-tetramers.

**BrdU incorporation assay**. To analyze proliferation ability upon LCMV challenge, indicated mice were i.p. injected with BrdU (1 mg/mice) 2 h before sacrifice. BrdU staining was performed with eBioscience BrdU Staining Kit for Flow Cytometry FITC (eBioscience) according to the manufacturer's protocol. In brief, splenocytes were stained with surface markers, then fixed and treated with DNase I. Cells were stained with fluorochrome conjugated anti-BrdU antibody and analyzed with flow cytometry. To analyze self-renewal capacity, indicated mice were i.p. injected with BrdU (1 mg/mice) once then treated with BrdU (0.8 mg/ml) in their drinking water for 7 days before sacrifice. BrdU solution was prepared in sterile water and protected from light. BrdU staining was performed as previously described.

**In vivo proliferation assay and MACS purification**. To analyze proliferation ability upon LCMV challenge, purified P14 CD5$^{lo}$, Ly6C$^-$, and Ly6C$^+$ naive CD8$^+$ T cells were labeled with CTV (2.5 μM), then transferred to host B6 mice infected with LCMV ($2 \times 10^5$ pfu/mice) 3 days before. After 36 h, spleens and LNs were pulled and enriched with CD8$^+$ T cells using CD8α$^+$ T Cell Isolation Kit (Miltenyi Biotec). First, single-cell suspensions of pulled spleen and LNs were suspended in 400 μl of MACS buffer (DPBS, pH 7.2, 0.5% bovine serum albumin, 2 mM EDTA). 100 μl of CD8α$^+$ T Cell Biotin-Antibody Cocktail were added and incubated for 10 min in ice. Stained cells were washed and resuspended in Anti-Biotin Microbeads containing buffer (Mixture of Anti-Biotin Microbeads 100 μl and MACS buffer 400 μl), then incubated for 10 min in ice. Cells were washed and resuspended in 500 μl of MACS buffer. LS Column (Miltenyi Biotec) was placed in QuadroMACS (Miltenyi Biotec) and washed with 2 ml of MACS buffer. Stained cells were applied to the column and washed twice with 2 ml of MACS buffer. The flow-through, now enriched with CD8$^+$ T cells, were stained with surface markers and analyzed with flow cytometry.

**Active caspase3 assay**. Single-cell suspensions of spleens of LCMV infected mice were cultured with FITC-VAD-FMK (Promega) for 30 min in 37 °C CO$_2$ incubator. Cells were harvested, washed twice with 5% FBS containing DPBS, and stained with surface markers.

**Metabolic assay**. Oxygen consumption rate (OCR) and extracellular acidification rate (ECAR) were assessed using a 96-well XF Extracellular flux analyzer (Agilent Technologies) and Seahorse Wave Controller Software (Agilent Technologies), according to the manufacturer's protocol. In brief, P14 CD5$^{lo}$, Ly6C$^-$, or Ly6C$^+$ cells were transferred to WT B6 mice, followed by LCMV infection ($2 \times 10^5$ pfu/mouse). At 7 dpi, donor cells were purified from the spleens. Purified donor cells were suspended in Seahorse assay medium, and seeded on Cell-Tak (Corning) (22.4 μg/ml, 25 μl/well) pre-treated Seahorse XF96 Cell Culture Microplate ($3 \times 10^5$ cells/well). The plate was centrifuged at $200 \times g$ for 1 min (zero-break), and transferred to CO$_2$ free 37 °C incubator for 30 min. Seahorse assay medium were added to top 175 μl in each well. After 25 min of additional incubation, cells were analyzed in Seahorse XF96 analyzer. For OCR analysis, Seahorse XF Cell Mito Stress Test Kit (Agilent Technologies) was used. Final concentrations of the inhibitors were as follows: Oligomycin 1.0 μM, FCCP 2.0 μM, Rotenone/antimycin A 0.5 μM. For ECAR analysis, Seahorse XF Glycolysis Stress Test Kit (Agilent Technologies) was used. Final concentrations of the inhibitors were as follows: glucose 10 mM, Oligomycin 1.0 μM, 2-DG 50 mM. Spare respiratory capacity (SRC) was calculated as difference between maximum OCR (After FCCP injection) and basal OCR. Glycolytic reserve (GR) was calculated as difference between maximum ECAR (After oligomycin) and glycolysis (after glucose). The acquired data were analyzed using Seahorse Wave Desktop (Agilent Technologies).

**Statistical analysis**. An unpaired two-tailed Student's *t*-test was performed to test statistical significance using Prism (GraphPad Software). Differences in mean values were considered statistically significant at a *P* value of less than 0.05.

**Reporting summary**. Further information on research design is available in the Nature Research Reporting Summary linked to this article.

## Data availability

RNA-Seq data and Microarray data have been deposited in the NCBI's Gene Expression Omnibus (GEO) database under the primary accession numbers GSE178445, GSE178447, GSE178448, and GSE178449. The authors declare that all data supporting the findings of this study are available within the paper and Supplementary information. Source data are provided with this paper.

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

## Acknowledgements

We thank K.S. Kim, S.H. Lim, and S.W. Lee (POSTECH) for providing GF and AF mice, and various transgenic and knock-out mice; S.J. Ha (Yonsei University) for LCMV virus; J.H. Lee (CNU) for helpful discussion and comments; M.J. Ryu and S.M. An (CNU) for administrative assistance; POSTECH and CNU flow cytometric core facilities for assistance with cell sorting; and G.S. Lee (POSTECH), M.S. Kim (CNU) for mice breeding and care. This work was supported by the Cooperative Research Program for Agriculture Science and Technology Development (Project No. PJ01336401 for C.H.Y.), and supported by National Research Foundation (NRF) funded by the Korean Ministry of Science and ICT (MSIT) (2018R1A2B2006793 for C.H.Y., and 2020R1A5A2031185 and 2020M3A9G3080281 for J.H.C.).

## Author contributions

Y.J.J., C.H.Y., and J.H.C. initiated and designed main idea of this study. Y.J.J. performed all major experiments; Y.J.J. and S.W.L. performed critical bioinformatic analysis from RNA-seq and microarray; G.W.L., Y.C.K., and H.O.K. performed some experiments for supplementary data; Y.J.J., S.W.L., C.H.Y., and J.H.C. analyzed and interpreted data and wrote the manuscript.

## Competing interests

The authors declare no competing interests.
