## [Peer Review File · Nature Communications]

Self-reactivity determines functional diversity of naive CD8+ T cells by co-opting tonic type I interferonREVIEWER COMMENTS

Reviewer #1 (Remarks to the Author):

The manuscript by Ju et al examine the mechanisms by which CD8 T cells which express different levels of CD5 also differ in their functional responses and potential for adopting different long-term fates. The current consensus is that differences in the behavior of CD5-hi vs CD5-lo T cells stem from their divergent abilities to sense endogenous ligands. The authors here show very intriguing data suggesting that the CD5-hi T cells (specifically Ly6C+ subsets) are poised to better perceive steady state type-I interferon signals, which in turn influence their activation and differentiation to SLECs vs MPECs. I have some questions regarding the data (and it's interpretation), as presented, which should be resolved before publishing this work. While the role of IFNAR-signaling is quite interesting, the authors should clarify the extent to which this mechanism contributes to the biology of CD5-hi T cells.

1. A conceptual concern is that the overall model being proposed is not all clear. How does self-reactivity regulate IFN sensitivity ? What underlies the heterogeneity within P14 cells – a monoclonal population of T cells which all bear the same self-reactivity ? Are T cells in different peripheral sites being exposed to different amounts of Type I IFNs (and other signals) leading to the differences ?
2. A key question that remains unresolved (see #1) is how the cd5-hi or ly6c+ cells gain more sensitivity to IFNs at the molecular level. The expression of IFNAR is an obvious explanation – but the data related to this (Ext Fig 2e) is very superficially handled. How many replicates were done of this experiment ? Is the tail of staining in the Ly6C+ cells indicative of higher IFNAR levels ? Do stat levels vary ? Have the authors looked at message for IFNAR ? Does antigen stimulation affect these levels differentially?
3. Figure 1.i : Given the role of tonic signaling in maintaining CD5 levels, does the level of CD5 drop in the cells transferred to the Tap- hosts ? Is the reduction in conversion to Ly6c+ cells secondary to this effect ? Do OT1 Ly6C- cells convert to Ly6C+ in a Tap- host ?
4. Figure 3.f. I am not sure I understand why the 2070 genes regulated by IFN are not affected in the Ly6C- vs C+ analysis ? Is the data suggesting that these 2070 genes are equally upregulated in both (if not, since all cells are being exposed to IFN, why would the cells only express 172 and not the other 2070). Please clarify.
5. While the influence of IFNAR signaling on Ly6C expression itself is quite convincing (Fig 3d), it is pretty evident that the ability of the cd5-hi or ly6c+ cells to respond to IL-2/12/18 is not so reliant on type 1 IFNs (Fig 4 b&d, the IFNAR- T cells still make a pretty strong response in this assay). Does this not suggest that the functional differences are not primarily driven by IFNAR ? Have the authors examined CD25/CD122 and IL-12Rb1/b2 expression in the Ly6C+ or CD5-hi cells ?
6. In page 15, the authors claim that "Together, these data suggest that the at least >7 days of T1IFN exposure is required". Was IFN blockade or IFNAR- cells used in this P14 transfer model to allow the authors to state that the 7+ day exposure was to IFNs ?

Reviewer #2 (Remarks to the Author):

In this study, Ju and colleagues examine the compelling issue of heterogeneity with the naive compartment of CD8+ T cells, and how heterogeneity is shaped by selection on self ligands and sensing of type I interferons. The central hypothesis throughout is the notion that type I interferon sensing is a key signal that promotes the emergence of "naïve" CD8+ T cells displaying a Ly6C+ CD183+ phenotype, and that these cells are preferentially responsive in inflammatory settings. This is a compelling area inquiry that builds on previous work in the field (e.g. References 7-15), and has potential to provide new insight into the pre-immune T cell repertoire, immunodominance, and other aspects of host defense. As detailed below, there are some key concerns with the study

in its current form:

Major Points:

1. In this study, the authors define "naïve" CD8+ T cells as those that are "low" for the activation marker CD44. However, this is set on a somewhat arbitrary gate of a marker that exhibits a continuous range of expression densities (Figure 1a) and may easily contain "contaminants" from the antigen-experienced compartment (e.g. virtual-memory (VM) CD8+ T cells that are CD44-hi CD122+ and express high densities of Eomes and other markers). Thus, all results downstream of this could be due to contamination with a small number of VM cells. For this reason, greater care and rigor are required to definitively define the "naïve" nature of these cells, and to ensure that truly naive populations are studied at high purity (i.e. using additional markers in addition to CD44-lo). Consistent with the idea of potential contamination within the CD44-lo gate, the production of IFN-gamma following in vitro treatment with IL-12 + IL-18 (Figure 4a) is an innate-like property that has been previously defined for VM cells. It is also possible that the CD44-lo Ly6C+ cells represent transitional intermediates that are on their way to becoming virtual memory cells. This notion is consistent with the data in Figure 2d, which shows that 25% of transferred Ly6C- cells upregulate Ly6C within 7 days following transfer into wild-type recipients. Do these Ly6C+ cells also acquire other markers of VM cells? In sum, a demonstration that naive CD8+ T cells displaying a Ly6C+ phenotype are preferentially poised to respond to infection or cytokine challenge would be a meaningful advance. However, more rigorous characterization and purification procedures are needed to demonstrate that the cells in question are truly naive, and are not CD44-hi CD122+ VM cell contaminants or a transitional population that is destined for the VM compartment.

2. In experiments in which naïve CD8+ T cells acquire the Ly6C-hi phenotype and other phenotypes (e.g. Figure 2d), do the cells remain naïve with respect to all other parameters? It is possible that cells are acquiring the Ly6C-hi phenotype because they are differentiating into activated cells or VM cells.

3. The dependency of Ly6C expression on type I interferons is an interesting finding, but the novelty is undercut by previous work demonstrating the same effect (e.g. References 16-17).

Minor Points:

4. In panels such as Figure 1b, analogous plots should also depict marker expression by CD44-hi CD8+ T cells, as a reference.

5. For the RNA-Seq in Figure 3, were cells first gated on CD44-lo? This is not mentioned in the text or figure legend.

6. Are Ly6C+ thymocytes recirculating mature T cells, or T cells that have recently developed?

7. How is the CD5-lo vs. CD5-hi gate drawn? CD5 expression density has a continuous distribution, and gating seems to be made arbitrarily.

8. In Figure 5a, what are the ratios of donor cell populations at endpoint if mice are not infected with LCMV? Is there a difference in survival and engraftment at baseline?

9. A useful positive control for many experiments would be CD44-hi CD122+ VM cells, to see how these compare to the Ly6C+ "naive" cells.

10. For Figure 5, what is the natural distribution of naive CD5-lo, Ly6C-, and Ly6C+ populations in P14 Rag1-/- mice?

Reviewer #3 (Remarks to the Author):

This paper describes the genetic and functional differences between CD5hi Ly6C- and Ly6C+ CD8

T cells. They find that development of the latter requires tonic exposure to type I IFNs, and results in cells that expand more rapidly after LCMV infection and are more likely to differentiate into short-lived effectors and less likely to become memory cells. The authors conclude that in addition to some degree of self-reactivity, exposure to steady-state levels of type I IFNs shape the functional response of CD8 T cells. Although the focus of this work is rather narrow, there are a great deal of data, often generated in adoptive transfer models in which the different subsets "compete" in the same environment, that support the conclusions.

Specific comments:

1. In Figs. 5a,b it is shown that Ly6C⁻ cells expand less well in response to LCMV than Ly6C⁺ cells. The authors say this is "surprising" but don't explain why they think so. I agree it is surprising, because LCMV rapidly induces type I IFN production, and as shown in vitro they induce Ly6C expression on previously Ly6C⁻ cells in at least 24 hrs, perhaps less. So unlike the steady-state experiments, one might have thought that during a viral response with much higher type I IFN levels there would be little difference between cells that were Ly6C⁻ vs. Ly6C⁺ at the time of transfer.
2. In panels 2f and 2g, how much IFN β was added? In panel h, what is the unit for the numbers .25, 1, and 5 -- concentration of IFN β , or time?
3. In Fig. 2e, there was only a difference at one concentration, 0.1 and 1, so it's not possible to say how different the responses really are with much certainty. A finer titration between those 2 points would be helpful (e.g. 2-fold). Also, units missing on the Y axis -- $\mu\text{g/ml}$? μM ? what?
4. What are T1IFN-induced donor-derived Ly6C⁺ cells (inLy6C⁺), I did not seem them defined as such? In the text it seems they are generated in vivo ("Ly6C⁻ P14 cells were transferred to and parked in B6 hosts for 7 and 21 days to generate Ly6C⁺ cells intermittently induced by T1IFN (inLy6C⁺)", but what does "intermittently" then refer to? Are they the cells described in the Methods section "In vitro Ly6C induction"? If so, they should be defined as inLy6C⁺ there, too, so there is no ambiguity.
5. It seems unlikely, but since the authors have the data: are the TCR levels different on Ly6C⁺ and Ly6C⁻ CD8 T cells?

Reply to the comments raised by Reviewers:

Reviewer #1 (Remarks to the Author):

The manuscript by Ju et al examine the mechanisms by which CD8 T cells which express different levels of CD5 also differ in their functional responses and potential for adopting different long-term fates. The current consensus is that differences in the behavior of CD5-hi vs CD5-lo T cells stem from their divergent abilities to sense endogenous ligands. The authors here show very intriguing data suggesting that the CD5-hi T cells (specifically Ly6C⁺ subsets) are poised to better perceive steady state type I interferon signals, which in turn influence their activation and differentiation to SLECs vs MPECs. I have some questions regarding the data (and its interpretation), as presented, which should be resolved before publishing this work. While the role of IFNAR-signaling is quite interesting, the authors should clarify the extent to which this mechanism contributes to the biology of CD5-hi T cells.

1. A conceptual concern is that the overall model being proposed is not all clear. How does self-reactivity regulate IFN sensitivity? What underlies the heterogeneity within P14 cells – a monoclonal population of T cells which all bear the same self-reactivity? Are T cells in different peripheral sites being exposed to different amounts of type I IFNs (and other signals) leading to the differences?

How does self-reactivity regulate IFN sensitivity? To address this question, we performed additional two adoptive transfer experiments. First, FACS-purified naive OT-I CD8⁺ T cells were injected i.v. to either WT or *Tap1*^{-/-} recipient hosts and, 3 days later, cells from spleen (SP) and lymph nodes (LN) were analyzed for STAT1 phosphorylation (pSTAT1) on OT-I donor cells upon brief (30 min) exposure to IFN-β in vitro. We found that T1IFN-induced pSTAT1 in OT-I cells is lower in *Tap1*^{-/-} hosts than WT hosts (**provided as the reviewer's perusal in Reviewer Fig. 1a, b**). Second, FACS-purified naive OT-I cells were injected i.v. to either B6 or congenically distinct OT-I hosts (to cause reduced self-reactivity due to robust intra-clonal competition for the same self-ligands between donor and recipient OT-I cells) and, 5 days later, cells from SP and LN were treated with IFN-β to analyze pSTAT1 by flow cytometry. We found that T1IFN-induced pSTAT1 in donor OT-I cells is significantly reduced in OT-I hosts (note that CD5 level is also decreased due to reduced TCR contacts with self-ligands) compared to control B6 hosts (**provided as the reviewer's perusal in Reviewer Fig. 1c-e**). Notably, despite the reduced pSTAT1, level of T1IFN receptor (IFNAR) was unchanged (**Reviewer Fig. 1b, e**). Hence, based on these data, we believe that a steady-state TCR-self signal positively regulates the sensitivity of naive CD8⁺ T cells to T1IFN. We now added the data from Reviewer Fig. 1c-e to **Supplementary Fig. 2k in the revised manuscript** and also mentioned this point in the Result section (**Page 10, lines 1-4**; highlighted with track change).

Reviewer Figure 1. TCR-self-MHC interaction is crucial for T1IFN responsiveness. FACS-purified naive OT-1 CD8⁺ T cells were transferred to (a-b) WT or *Tap1*^{-/-} hosts, or (c-e) B6 or OT-1 hosts. After 3-5 days, the cells from spleen and lymph nodes were harvested and MACS-enriched for CD8⁺ cells. Enriched CD8⁺ cells were treated with IFN- β (10 ng/ml) for 30 minutes. The cells were fixed right away and permeabilized with MeOH, followed by staining with fluorochrome conjugated antibodies including anti-pSTAT1, anti-IFNAR1, and anti-CD5.

Although positive correlation between self-reactivity (measured by the level of CD5 expression on T cells, i.e., CD5^{lo} versus CD5^{hi}) and cytokine responsiveness (for γ c cytokines such as IL-2, IL-7 and IL-15) has been well demonstrated in previous literatures, the exact mechanisms underlying such correlation remain unclear. Some studies invoke different modulation of the cytokine receptor expression (Palmer et al, Immunol. Cell Biol., 2011; White et al, Nat. Comm., 2016; Cho et al, Nat. Comm., 2016), but others suggest different intrinsic signaling capability (Persaud et al, Nat. Immunol., 2014; Fulton et al, Nat. Immunol., 2015).

In this regard, we found that the level of IFNAR1 and IFNAR2 on naive CD8⁺ T cells are not different between CD5^{lo} and CD5^{hi} cells, albeit the latter subset (both Ly6C⁻ and Ly6C⁺ cells) shows moderately increased level of intracellular signaling molecules, STAT1 and STAT2, (provided as the reviewer's perusal in Reviewer Fig. 2). Therefore, we think that a signal

from TCR contacts with self-ligands positively regulates T1IFN response presumably through shaping cell-intrinsic signaling ability downstream of IFNAR and not via different modulation in IFNAR expression. We now added the data from Reviewer Fig. 2 to **Supplementary Fig. 2k in the revised manuscript**.

Reviewer Figure 2. Naive CD5^{hi} CD8⁺ T cells express comparable level of IFNAR1 and IFNAR2, and moderately increased levels of STAT1 and STAT2 compared to naive CD5^{lo} CD8⁺ T cells. Ex vivo splenocytes of WT B6 were stained for (a) IFNAR1, (b) IFNAR2, (c) total STAT1, and (d) total STAT2 and analyzed by flow cytometry.

We also provide new data showing a marked reduction in the ability to produce IFN- γ in response to IL-12 and IL-18 when P14 naive CD8⁺ T cells were adoptively transferred to *Tap1*^{-/-} mice (**provided as the reviewer's perusal in Reviewer Fig. 3a, b**). Moreover, the P14 cells recovered from *Tap1*^{-/-} hosts showed a significant (~2-3-fold) reduction in antigen-specific expansion when reinjected to B6 mice and then infected with LCMV (**provided as the reviewer's perusal in Reviewer Fig. 3c**). Hence, based on these data, we further highlight the importance of TCR-self interaction in shaping the functional response of naive CD8⁺ T cells. We now added the data from Reviewer Fig. 3a, b to **Supplementary Fig. 4d in the revised manuscript** and mentioned this point in the Result section (**Page 13, lines 17-20**)

Reviewer Figure 3. Self-recognition is crucial for cytokine responsiveness. P14 CD8⁺ T cells were transferred to WT or *Tap1*^{-/-} hosts. After 5 days, (a-b) splenocytes were harvested and cultured with IL12/18 (± IL2) for 18 hours. Golgiplug were added for 5 hours, then fixed and permeabilized for intracellular staining of IFN-γ. P14 donor cells producing IFN-γ were analyzed with flow cytometry. (c) Naïve P14 donor cells were purified from the LN, then transferred to WT B6 hosts, followed by LCMV infection next day. After 9 DPI, donor recovery was analyzed from the spleen and LN.

What underlies the heterogeneity within P14 cells – a monoclonal population of T cells which all bear the same self-reactivity? It has been generally accepted that the nature of determining intrinsic self-reactivity among individual naïve CD8⁺ T cells is far more complex and promiscuous for both polyclonal and even monoclonal T cell population.

For example, a study of Haluszczak et al (JEM, 206:435, 2009) showed phenotypic and functional difference within a pool of the same antigenic-specific CD8⁺ T cells in unimmunized normal B6 mice (using MHC tetramers loaded with specific peptide antigens); here, ~10-40% of total CD8⁺ T cells with specificity to the same antigens had characteristics of memory cells (known as ‘virtual memory’) with enhanced functional response compared to their naïve counterparts, suggesting heterogeneity even in the same T cell clones. Furthermore, in close accord with our findings with CD5^{lo} vs. CD5^{hi} subsets of P14 cells, distinct functional characteristics (i.e., proliferative responses to either IL-2 or IL-15 in vitro) have previously been demonstrated in other TCR transgenics (e.g., 2C and HY; Cho et al, Immunity, 2010). Hence, we believe that the relative influence of self-reactivity of a given TCR clonotype is not identical but rather variable from one subset to another even in monoclonal CD8⁺ T cell population.

Nevertheless, we do agree more direct evidence is needed for the effect of different self-reactivity in P14 cells. We thus performed adoptive transfer experiments of FACS-purified P14 naïve CD5^{lo} Ly6C⁻, CD5^{hi} Ly6C⁻ and CD5^{hi} Ly6C⁺ cells into B6 hosts to address whether these subsets indeed variably respond to T1IFN in vivo in a manner dependent on the relative difference in their self-reactivity. For this, we evaluated the ability of Ly6C

induction in these cells, given that Ly6C expression depends on T1IFN signal and is largely affected by self-reactivity (**previous data in Fig. 2c-j**). At one week after transfer, a proportion of P14 CD5^{lo} Ly6C⁻ cells and CD5^{hi} Ly6C⁻ cells become Ly6C⁺ cells, with the latter being much higher than the former (42.9% vs. 21.5%; **provided as the reviewer's perusal in Reviewer Fig. 4a, b**). This finding indicates different sensitivity of these two monoclonal subsets to T1IFN.

Moreover, we also found that a small but significant fraction of P14 CD5^{hi} Ly6C⁺ cells (which express uniformly high levels of Ly6C) appear to downregulate Ly6C and return to Ly6C-negative phenotype (**provided as the reviewer's perusal in Reviewer Fig. 4a, b**), which reflects at least some competition even in P14 CD5^{hi} Ly6C⁺ donor cells for T1IFN. So, to further confirm such intraclonal competition, we adoptively transferred FACS-purified Ly6C⁻ P14 cells into congenically distinct P14 hosts. A proportion of Ly6C⁺ cells derived from donor Ly6C⁻ cells was nearly identical with that of host Ly6C⁺ cells (**provided as the reviewer's perusal in Reviewer Figure 4c**). Therefore, we conclude that heterologous responses apply not only for polyclonal but also for the same monoclonal CD8⁺ T cells, and that the effect is at least in part a reflection of in vivo competition for limiting amounts of homeostatic cues, T1IFN and self-ligands herein.

Reviewer Figure 4. Monoclonal P14 CD8⁺ T cells variably respond to in vivo T1IFN. (a-b) Naive P14 CD8⁺ T cell subsets (CD5^{lo} Ly6C⁻, CD5^{hi} Ly6C⁻, CD5^{hi} Ly6C⁺) were FACS-purified, then transferred to WT B6 hosts. After 5 days, proportion of Ly6C⁺ cells in the donor cells was analyzed from the spleen and LN. (c) FACS-purified P14 Ly6C⁻ CD8⁺ T cells were transferred to congenically different P14 hosts. After 7 days, proportion of Ly6C⁺ cells in the donor and the host P14 CD8⁺ T cells was analyzed with flow cytometry

Are T cells in different peripheral sites being exposed to different amounts of type I IFNs (and other signals) leading to the differences? We appreciate the reviewer for the valuable comments. Since naive T cells preferentially home to secondary lymphoid organs (SLO), it is possible that these cells in different SLO may have access to different levels of T1IFN, contributing to heterogeneity. To address this, we examined the proportion of Ly6C⁺ cells (as

a measurer of in vivo T1IFN response) in naïve CD8⁺ T cells from different SLO, including spleen and various LNs (inguinal, axillary, cervical and mesenteric). We found nearly identical frequency of Ly6C⁺ cells across these SLOs (~25-29%; **provided as the reviewer's perusal in Reviewer Fig. 5a**).

In addition, we also found that naïve CD8⁺ T cells from different SLOs have comparable IFN- γ -producing ability upon in vitro culture with IL-12/IL-18 (~7-10%; **provided as the reviewer's perusal in Reviewer Fig. 5b**). So, we think it is unlikely that an in vivo concentration of T1IFN varies considerably depending on different SLOs and that the observed functional differences are related to differential tissue homing of T cells.

Reviewer Figure 5. Naïve CD8⁺ T cells from different SLOs have nearly identical percentage of Ly6C⁺ cells and the IFN- γ production ability in response to IL2/12/18. Cells from indicated SLOs were harvested and (a) proportion of naïve CD8⁺ T cell subsets were analyzed. (b) Cells from the indicated SLOs were cultured with IL2/12/18 for 18 hours. Golgiplug was added for 5 hours, then fixed and permeabilized for intracellular staining of IFN- γ . IFN- γ production of naïve CD8⁺ T cells was analyzed with flow cytometry.

2. A key question that remains unresolved (see #1) is how the CD5^{hi} or Ly6C⁺ cells gain more sensitivity to IFNs at the molecular level. The expression of IFNAR is an obvious explanation – but the data related to this (Ext Fig 2e) is very superficially handled. How many replicates were done of this experiment? Is the tail of staining in the Ly6C⁺ cells indicative of higher IFNAR levels? Do stat levels vary? Have the authors looked at message for IFNAR? Does antigen stimulation affect these levels differentially?

With regard to the reviewer's concerns, we confirmed that CD5^{lo} Ly6C⁻, CD5^{hi} Ly6C⁻, and CD5^{hi} Ly6C⁺ cells show comparable levels of IFNAR1 and IFNAR2 (see Reviewer Fig. 2). Similar results were also observed at the mRNA levels, with no difference in *Ifnar1*, *Ifnar2*, *Stat1* and *Stat2* gene expressions (**provided as the reviewer's perusal in Reviewer Fig. 6**).

Reviewer Figure 6. CD8⁺ T cell subsets have comparable levels of mRNA transcripts coding IFNAR1, IFNAR2, STAT1, and STAT2. Naïve CD8⁺ T subsets (CD5^{lo} Ly6C⁻, CD5^{hi} Ly6C⁻, CD5^{hi} Ly6C⁺) from WT B6 mice were used for RNA-seq. FPKM of (a) Ifnar1, (b) Ifnar2, (c) Stat1, and (d) Stat2 was analyzed.

With regard to the effect of antigenic stimulation on IFNAR expression, we performed an experiment in which naïve CD8⁺ T cells were stimulated with anti-CD3 and -CD28 antibodies in vitro. We found significant increase in both IFNAR1 and IFNAR2 expression on T cells after 2 d (but not 1 d) TCR stimulation (**provided as the reviewer's perusal in Reviewer Fig. 7a, b**). Since we could detect CD44 up-regulation, but not proliferation, from day 1 (**provided as the reviewer's perusal in Reviewer Fig. 7c, d**), it is likely that the enhanced IFNAR expression depends on relatively longer and stronger TCR signaling. Therefore, we believe that differential T1IFN sensitivity among CD5^{lo} Ly6C⁻, CD5^{hi} Ly6C⁻ and CD5^{hi} Ly6C⁺ cells is not due to different IFNAR levels but instead reflects different intrinsic signaling capacity positively regulated by self-reactivity (**see above our reply to reviewer's question 1**).

Reviewer Figure 7. TCR stimulation leads to significant increase in the expression of both IFNAR1 and IFNAR2 on T cells activated for 2 days but not 1 day. FACS-purified naïve CD8⁺ T cells were CTV-labelled, then divided into 3 groups. The first group (NS) was cultured without

stimulation for 2 days. The second group (1d activated) was cultured without stimulation at the first day, then activated with anti-CD3 and anti-CD28 at the second day. The third group (2d activated) was activated with anti-CD3 and anti-CD28 for 2 days. For the survival of T cells, all samples were subjected with low dose of IL-7 (1 ng/ml). Then, all groups were analyzed for expression of (a) IFNAR1, (b) IFNAR2, (c) CD44, and (d) CTV dilution with flow cytometry.

3. Figure 1.i: Given the role of tonic signaling in maintaining CD5 levels, does the level of CD5 drop in the cells transferred to the *Tap1*^{-/-} hosts? Is the reduction in conversion to Ly6C⁺ cells secondary to this effect? Do OT1 Ly6C⁻ cells convert to Ly6C⁺ in a *Tap1*^{-/-} host?

It has been well accepted that the level of CD5 on T cells is associated with the relative strength of tonic self-TCR signaling and thus a number of previous studies have already shown that MFI for CD5 level on T cells is decreased when adoptively transferred to *Tap1*-deficient hosts (Cho et al, Immunity, 2010; Takada et al, J. Exp. Med., 2009; **also provided here as the reviewer’s perusal in Reviewer Fig. 8a**).

To address the reviewer’s concern that the decreased conversion from Ly6C⁻ cells to Ly6C⁺ cells in *Tap1*^{-/-} mice (**previous data in Fig. 2i**) might be secondary to the reduced CD5 expression, we performed adoptive transfer experiments of FACS-purified OT-I Ly6C⁻ cells to either WT or *Tap1*^{-/-} hosts. In line with our previous data with B6 Ly6C⁻ cells, OT-I Ly6C⁻ cells led to significantly decreased conversion to Ly6C⁺ cells in *Tap1*^{-/-} hosts compared to WT hosts (**provided as the reviewer’s perusal in Reviewer Fig. 8b**). As expected, CD5 levels on donor OT-I cells were also decreased in *Tap1*^{-/-} hosts, although still considerably high relative to B6 naive CD8⁺ T cells (**provided as the reviewer’s perusal in Reviewer Fig. 8c**).

Hence, these data further support our previous conclusion that tonic self-TCR signaling provides the uniform positive influence on T11FN-induced Ly6C⁺ cell generation for both polyclonal and monoclonal naive CD8⁺ T cells. We now added the new data from Reviewer Fig. 8b to **Supplementary Fig. 2j in the revised manuscript**.

Reviewer Figure 8. OT-I Ly6C⁻ cells show significantly reduced ability to convert to Ly6C⁺ cells after transfer to *Tap1*^{-/-} hosts compared to B6 hosts. (a) Ly5.1 CD8⁺ T cells were transferred to B6 WT or *Tap1*^{-/-} hosts. After 3 days, splenocytes were harvested, and then CD5 expression of the donor cells was analyzed with flow cytometry. (b-c) FACS-purified naive OT-I Ly6C⁻ CD8⁺ T cells were transferred to B6 WT or *Tap1*^{-/-} hosts. After 5 days, (b) proportion of Ly6C⁺ cells (c) and CD5 expression in the donor OT-I cells were analyzed by flow cytometry.

4. Figure 3.f: I am not sure I understand why the 2070 genes regulated by IFN are not affected in the Ly6C- vs Ly6C+ analysis? Is the data suggesting that these 2070 genes are equally upregulated in both (if not, since all cells are being exposed to IFN, why would the cells only express 172 and not the other 2070). Please clarify.

It should be emphasized that, in our RNA-seq experiment (previous data in Fig. 3f), the cells were treated for 24 h with relatively high concentrations of IFN- β (10 ng/ml). Although such supraphysiological level of IFN- β (relative to a trace amount of T1IFN in vivo) is useful to obtain as much information as possible for a full range of T1IFN-induced transcriptome, we also expect that the induced gene profiles might not be exactly the same as those induced by in vivo T1IFN. In fact, when we treated naive CD8⁺ T cells with titrated doses of IFN- β (0, 0.1, and 1 ng/ml), we found that down-regulation of *Itgae* and *Ifngr1* genes is observed with relatively high doses of IFN- β (>1 ng/ml), whereas up-regulation of *Ccl5* gene with a lower dose (<0.1 ng/ml) (provided as the reviewer's perusal in Reviewer Fig. 9a-c). So, we think that the observed difference in the number of genes shared is presumably a reflection of difference in the given effective concentration of T1IFN between in vitro and in vivo.

Reviewer Figure 9. Different genes require different dose of IFN β to change the gene expression. FACS-purified naive CD8⁺ T cells were treated with different dose of IFN- β (0, 0.1, 1 ng/ml) for 16 hours. Then, the cells were harvested and used to obtain RNA. cDNA was prepared from the RNA, then used for quantitative PCR with primers of (a) *Itgae*, (b) *Ifngr1*, and (c) *CCL5* (and 18s rRNA for house keeping gene).

We understand, however, that the above dosage effect would not completely rule out the possibility that the shared 172 genes are — at least some if not all — irrelevant to tonic T1IFN signaling. Given the fact that the core enrichment genes associated with cell division and proliferation overlap with a number of IFN- β -regulated genes (Amgen cluster III; previous data in Table 1), we performed additional RNA-seq experiment with *Ifnar*^{-/-} CD5^{lo} and CD5^{hi} naive CD8⁺ T cells to address whether the cluster III genes are indeed regulated in a T1IFN-dependent manner. We found that while skewing clearly toward WT CD5^{hi} cells over CD5^{lo} cells, such skewing is markedly reduced in comparison between *Ifnar*^{-/-} counterparts (provided as the reviewer's perusal in Reviewer Fig. 10). So, these data clearly indicate that a significant fraction of cluster III genes are indeed regulated in a manner dependent on a physiological level of T1IFN signal.

Reviewer Figure 10. While genes in Cluster III was enriched in WT CD5^{hi} cells compared to WT CD5^{lo} cells, the enrichment was reduced between Ifnar^{-/-} CD5^{hi} and CD5^{lo} cells. The RNA-seq data from CD5^{lo} and CD5^{hi} cells from either (a) WT or (b) Ifnar^{-/-} mice were used for gene set enrichment analysis (GSEA) with Amgene Cluster III.

5. While the influence of IFNAR signaling on Ly6C expression itself is quite convincing (Fig 3d), it is pretty evident that the ability of the CD5-hi or Ly6C+ cells to respond to IL-2/12/18 is not so reliant on type I IFNs (Fig 4 b&d, the IFNAR- T cells still make a pretty strong response in this assay). Does this not suggest that the functional differences are not primarily driven by IFNAR? Have the authors examined CD25/CD122 and IL-12Rb1/b2 expression in the Ly6C+ or CD5-hi cells?

We appreciate the reviewer for the valuable comments. However, we should emphasize that IL-12/IL-18-driven innate response was relatively weak for naive CD8⁺ T cells (~0.5-6%; **previous data in Fig. 4a, b**). So, we additionally treated IL-2 just as a control to validate functional relevance of the IL-12/IL-18-driven innate response, as the addition of IL-2 was known to substantially increase the magnitude of this response in T cells (~2-40%; **previous data in Fig. 4a, b**).

With regard to IL-2R and IL-12R expressions, we could not detect any significant differences in the expression of CD25, CD122, IL-12Rβ1 and IL-12Rβ2 (**provided as the reviewer's perusal in Reviewer Fig. 11**). So, we think it is unlikely that the differential innate responses among naive CD8⁺ T cell subsets, especially Ly6C⁺ cells, result from different levels of these cytokine receptors.

Reviewer Figure 11. Naive CD8⁺ T cell subsets have comparable receptor expression for IL-2 and IL-12. Expressions of the (a) CD25, (b) CD122, (c) CD212, and (d) IL12Rβ2 were analyzed in the naïve CD8⁺ T subsets (CD5^{lo}, CD5^{hi} Ly6C⁻, CD5^{hi} Ly6C⁺) and MP from the spleen of WT B6 mice.

6. In page 15, the authors claim that “Together, these data suggest that the at least >7 days of T1IFN exposure is required”. Was IFN blockade or IFNAR^{-/-} cells used in this P14 transfer model to allow the authors to state that the 7 day exposure was to IFNs?

In the paper, we wanted to address whether the enhanced antigen-specific expansion capacity of Ly6C⁺ cells depends on the duration of in vivo T1IFN exposure for the generation and maintenance of these cells. We believe that the in vivo parking experiments (**previous data in Fig. 5f**) and our blocking experiments with anti-IFNAR (**previous data in Fig. 4d, 7e-g**) provide a reasonable support, though not proof, for a potential role of in vivo exposure of tonic T1IFN, as this cytokine was crucial for generating Ly6C⁺ cells and modulating their innate response (**previous data in Fig. 2d, 4c**). However, we do agree that specifying “7-day exposure” may cause confusion, and therefore we removed it in the revised manuscript (**Page 16, lines 17-18**)

Reviewer #2 (Remarks to the Author):

In this study, Ju and colleagues examine the compelling issue of heterogeneity with the naive compartment of CD8⁺ T cells, and how heterogeneity is shaped by selection on self ligands and sensing of type I interferons. The central hypothesis throughout is the notion that type I interferon sensing is a key signal that promotes the emergence of “naive” CD8⁺ T cells displaying a Ly6C⁺ CD183⁺ phenotype, and that these cells are preferentially responsive in inflammatory settings. This is a compelling area inquiry that builds on previous work in the field (e.g. References 7-15), and has potential to provide new insight into the pre-immune T cell repertoire, immunodominance, and other aspects of host defense. As detailed below, there are some key concerns with the study in its current form:

Major Points:

1. In this study, the authors define “naive” CD8⁺ T cells as those that are “low” for the activation marker CD44. However, this is set on a somewhat arbitrary gate of a marker that exhibits a continuous range of expression densities (Figure 1a) and may easily contain “contaminants” from the antigen-experienced compartment (e.g. virtual-memory (VM) CD8⁺ T cells that are CD44^{hi} CD122⁺ and express high densities of Eomes and other markers). Thus, all results downstream of this could be due to contamination with a small number of VM cells. For this reason, greater care and rigor are required to definitively define the “naive” nature of these cells, and to ensure that truly naive populations are studied at high purity (i.e. using additional markers in addition to CD44^{lo}). Consistent with the idea of potential contamination within the CD44^{lo} gate, the production of IFN-gamma following *in vitro* treatment with IL-12 + IL-18 (Figure 4a) is an innate-like property that has been previously defined for VM cells. It is also possible that the CD44^{lo} Ly6C⁺ cells represent transitional intermediates that are on their way to becoming virtual memory cells. This notion is consistent with the data in Figure 2d, which shows that 25% of transferred Ly6C⁻ cells upregulate Ly6C within 7 days following transfer into wild-type recipients. Do these Ly6C⁺ cells also acquire other markers of VM cells? In sum, a demonstration that naive CD8⁺ T cells displaying a Ly6C⁺ phenotype are preferentially poised to respond to infection or cytokine challenge would be a meaningful advance. However, more rigorous characterization and purification procedures are needed to demonstrate that the cells in question are truly naive, and are not CD44^{hi} CD122⁺ VM cell contaminants or a transitional population that is destined for the VM compartment.

To address the reviewer’s overall concerns about the potential contamination of memory-phenotype (MP; and also VM) cells in our study, we performed several additional experiments. The new data and our previous data can be summarized as follows:

1) In our experiments with FACS-purified naive CD8⁺ T cell subsets (from B6 mice), CD44^{lo} CD62L^{hi} cells were defined as a naive and gated stringently for cell sorting to clearly separate them from CD44^{hi} CD62L^{lo/hi} cells with an average of > 99% purity. FACS data before and after cell sorting are **provided for the reviewer’s perusal in Reviewer Fig. 12, and also added in Supplementary Fig. 3a in the revised manuscript.**

Reviewer Figure 12. Sorting purity of naive CD8⁺ T cells subsets. Lymph nodes of WT B6 mice were harvested and prepped into single cells, then stained with fluorochrome conjugated α CD8, α CD44, α CD62L, α CD5, and α Ly6C for FACS-sorting. After every sort, sorting purities were ensured.

2) For other surface markers, MP (and also VM) cells have been well demonstrated to clearly differ from naive CD8⁺ T cells particularly in their high expression of CD122 (CD122^{hi}). In our experiment, the level of CD122 was lower in all three naive CD8⁺ T cell subsets (although slightly higher in CD5^{hi} cells than CD5^{lo} cells) than MP cells. Moreover, despite a fraction of naive cells showed expression of Ly6C and CD183, their MFIs were much lower than those of MP cells showing uniformly high density of these markers (**provided as the reviewer's perusal in Reviewer Fig. 13a-d, and now added to Supplementary Fig. 1b in the revised manuscript**).

Reviewer Figure 13. CD44^{lo} CD5^{hi} Ly6C⁺ cells show significantly lower expression of CD44, CD122, CD183, and Ly6C than those on MP cells. Expressions of the (a) CD44, (b) CD122, (c)

Ly6C, and (d) CD183 were analyzed in the naïve CD8⁺ T subsets (CD5^{lo}, CD5^{hi} Ly6C⁻, CD5^{hi} Ly6C⁺) and MP from the spleen of WT B6 mice.

3) In agreement with our data (**new data in Supplementary Fig. 1c in the revised manuscript**) and data from other reports (Cho et al, PNAS, 1999; Veiga-Fernandes et al, Nat. Immunol., 2000; Kersh et al, J. Immunol., 2003; DiSpirito et al, Cell Res, 2010), MP (and also VM) cells have already been well known to induce faster and stronger activation than naive cells upon antigen/TCR stimulation. Similarly, we also performed additional in vitro experiments in which FACS-purified CD44^{hi} MP CD8⁺ T cells and three naive (CD44^{lo}) CD8⁺ T cell subsets (i.e., CD5^{lo} Ly6C⁻, CD5^{hi} Ly6C⁻ and CD5^{hi} Ly6C⁺) were stimulated for 5 hr with plate-bound anti-CD3/CD28 mAbs and then analyzed for their early activation gene expression profiles using RNA-seq, and then compared with publicly available dataset for memory CD8⁺ T cells (GSE10239). All three naive CD8⁺ T cell subsets (including Ly6C⁺ cells) closely resembled each other and were distinctly different from CD44^{hi} MP cells (**provided as the reviewer's perusal in Reviewer Fig. 14a, b**). Notably, the levels of some early effector genes, such as *Il2*, *Ifng* and *Gzmb*, were still very low in all three naive subsets analyzed at this early time point, compared to those observed in MP cells (**provided as the reviewer's perusal in Reviewer Fig. 14c**).

Reviewer Figure 14. Expression of memory-related gene set in naive CD8⁺ T cell subsets was closely resembling to each other and is distinctly different from MP cells. Naïve CD8⁺ T subsets (CD5^{lo}, CD5^{hi} Ly6C⁻, CD5^{hi} Ly6C⁺) and CD44^{hi} MP cells were FACS-purified, then stimulated with plate-bound anti-CD3/CD28 mAbs for 5 hours. Stimulated cells were used for RNA-seq. (a-b) Publicly available data set for memory CD8⁺ T cells (Genes upregulated in memory CD8⁺ T cells compared to naive CD8⁺ T cells; GSE10239) were used to analyze the RNA-seq data. (a) Genes in the GSE10239 gene set were collected from the RNA-seq data and used to make the heat map. (b) Gene set enrichment analysis (GSEA) were carried out with Broad Institute GSEA software using the GSE10239 gene set and the RNA-seq data. (c) Several surface markers and early effector genes including Ifng, Il2, and Gzmb were analyzed.

4) As the reviewer points out, we agree that for CD8⁺ T cells, the innate-like property (i.e., IFN- γ production in response to IL-12/IL-18) is a major characteristic for MP (and also VM) cells rather than naive cells. Our data on total splenocytes cultured with IL-12/IL-18 (\pm IL-2; **previous data in Fig. 4**) showed, however, that such innate response applies to naive CD8⁺ T cells, with the highest response with CD5^{hi}Ly6C⁺ subset. Nevertheless, it is important to emphasize that the magnitude of such response in naive CD8⁺ T cells remained considerably lower than that of MP cells (~2.5-32% vs. 85% for IFN- γ ⁺ cells; **provided as the reviewer's perusal in Reviewer Fig. 15a, and also added to Supplementary Fig. 4a in the revised manuscript**). Moreover, such weaker response of naive cells (relative to MP cells) was also observed in the MFI level for IFN- γ expression on a per-cell basis (**provided as the reviewer's perusal in Reviewer Fig. 15b**).

The above data were further validated with a more stringent cell sorting strategy in which cells with extremely low level of CD44 (CD44^{exlo}; relative to CD44^{lo} used in our previous data) were sorted to completely avoid any potential contamination of MP (and VM) cells. Consistent with our data (**previous data in Fig. 4a**), additional new data with CD44^{exlo} naive CD8⁺ T cell subsets showed that the innate response (IL-12/IL-18-induced IFN- γ production) was greater for CD5^{hi}Ly6C⁺ cells than for CD5^{lo} and even CD5^{hi}Ly6C⁻ cells (~1.6-4.2% vs. 20%; (**provided as the reviewer's perusal in Reviewer Fig. 15c**).

Reviewer Figure 15. Naïve CD5^{hi} Ly6C⁺ CD8⁺ T cells have clearly distinct IFN- γ production ability compared to MP cells. (a-b) Splenocytes of WT B6 mice were culture with IL2/12/18 for overnight, followed by addition of Golgiplug for 5 hours. Cells were fixed and permeabilized for staining of IFN- γ . (a) Proportion of IFN- γ ⁺ cells and (b) IFN- γ MFI of IFN- γ ⁺ cells in naïve CD8⁺ T

subsets ($CD5^{lo}$, $CD5^{hi}$ Ly6C⁻, $CD5^{hi}$ Ly6C⁺) and $CD44^{hi}$ MP cells were analyzed by flow cytometry. (c) $CD8^{+}$ cells were gated with extremely low level of $CD44$ ($CD44^{exlo}$), then gated to three subsets of naive $CD8^{+}$ T cells ($CD5^{lo}$, $CD5^{hi}$ Ly6C⁻, $CD5^{hi}$ Ly6C⁺) to analyze IFN- γ production.

5) We compared the proliferative response of FACS-purified MP cells and three naive $CD8^{+}$ T cell subsets in response to either TCR stimulation or γ c cytokines (IL-2, IL-7 and IL-15) (provided as the reviewer's perusal in Reviewer Fig. 16, and also added to Supplementary Fig. 1c in the revised manuscript). MP cells showed markedly enhanced proliferative response when cultured with IL-2 or IL-15, but in contrast, all three naive subsets (even for Ly6C⁺ cells) failed to do so. Likewise, such weaker response of naive cells (relative to MP cells) was also apparent when cultured with IL-7, although a moderate proliferation in naive cells (particularly $CD5^{hi}$ subsets) was observed. As with cytokine stimulation, TCR stimulation with anti-CD3/CD28 showed much greater proliferation in MP cells than naive cells; note that we have previously shown that the enhanced TCR-induced proliferation of MP cells is attributed to higher production of and sensitivity to IL-2 (Cho et al, Nat. Comm., 2016).

Reviewer Figure 16. CD44^{hi} MP CD8⁺ cells have more pronounced proliferation capacity than all naive CD8⁺ T cell subsets. FACS-purified naive CD8⁺ T cell subsets (CD5^{lo}, CD5^{hi} Ly6C⁻, CD5^{hi} Ly6C⁺) and CD44^{hi} MP CD8⁺ T cells were labelled with CTV, then activated with indicated stimulation. CTV dilutions were analyzed with flow cytometry.

6) In agreement with the afore-mentioned higher proliferation of MP cells upon TCR stimulation, we also provide new data showing that IFN- γ production on CD8⁺ T cell subsets stimulated for 5 hr with PMA/ionomycin was much greater in CD44^{hi} MP cells than in CD44^{lo} naive cells for the percentages and MFIs (~5-40% vs. ~80% for IFN- γ ⁺ cells; ~600-1000 vs. ~4500 for IFN- γ MFI; **provided as the reviewer's perusal in Reviewer Fig. 17, and also added to Supplementary Fig. 1d in the revised manuscript**).

Reviewer Figure 17. CD44^{hi} MP CD8⁺ T cells have significantly higher IFN- γ production with regards to percentage and MFI in response to TCR stimulation compared to naive CD8⁺ T cell subsets. Splenocytes of WT B6 mice were cultured with PMA/ionomycin (plus golgi inhibitor) for 5 hours. (a) Proportion of IFN- γ ⁺ cells and (b) IFN- γ MFI of IFN- γ ⁺ cells in naive CD8⁺ T subsets (CD5^{lo}, CD5^{hi} Ly6C⁻, CD5^{hi} Ly6C⁺) and CD44^{hi} MP CD8⁺ T cells were analyzed by flow cytometry.

7) Besides our data described above, evidence on the “naive” nature of Ly6C⁺ cells (both for phenotype and function) dose not simply hinge on the use of polyclonal B6 CD8⁺ T cells. In this study, we provided extensive control data with monoclonal P14 cells. As the P14 mice used were all on a *Rag1*^{-/-} background, CD44^{hi} P14 cells were almost undetectable in these mice (**provided as the reviewer's perusal in Reviewer Fig. 18**). Therefore, we think that a potential contamination of MP (and VM) cells might be negligible (particularly after our stringent sorting procedure).

Reviewer Figure 18. CD44^{hi} MP CD8⁺ T cells were almost undetectable in P14 Rag1^{-/-} mice. CD44^{hi} cells were analyzed in CD8⁺ T cells of P14 WT mice and P14 Rag1^{-/-} mice

Collectively, we believe that all three CD8⁺ T cell subsets analyzed in our study are truly naive cells (showing variable functional capacity with each subset, especially Ly6C⁺ cells) that are distinctly different from MP (and VM) cells.

2. In experiments in which naive CD8⁺ T cells acquire the Ly6C-hi phenotype and other phenotypes (e.g. Figure 2d), do the cells remain naive with respect to all other parameters? It is possible that cells are acquiring the Ly6C-hi phenotype because they are differentiating into activated cells or VM cells.

We appreciate the reviewer for the valuable comments. However, in our hands (**previous data in Fig. 2d**), we could not find any evidence showing the possible transition of naive Ly6C⁺ cells to activated (or VM) cells. Instead, we found that Ly6C⁺ cells newly induced from adoptively transferred Ly6C⁻ cells still remained naive cell phenotype, CD44^{lo}CD183⁻ (**provided as the reviewer's perusal in Reviewer Fig. 19, and also added to Supplementary Fig. 2e in the revised manuscript**), which is in sharp contrast to MP (and VM) cells showing CD44^{hi}CD183⁺ phenotype.

Reviewer Figure 19. Both WT and Ifnar1^{-/-} naive CD8⁺ T cells maintain low level of CD44 expression at 1 week after the transfer. FACS-purified naive CD8⁺ T cells from WT B6 and Ifnar1^{-/-} mice were transferred to B6 hosts. After 1 week, CD44 expression of the donor cells was analyzed by flow cytometry.

Since analysis of donor cells at day 7 post-transfer might be too early to evaluate a possible transition of naive Ly6C⁺ cells to activated (or VM) cells, we also performed additional transfer experiments in which FACS-purified CD5^{lo} Ly6C⁻, CD5^{hi} Ly6C⁻ and CD5^{hi} Ly6C⁺ cells were transferred to B6 mice and analyzed 8 weeks post-transfer. Despite such longer period of time, nearly all of each donor subset (including CD5^{hi}Ly6C⁺ cells) remained CD44^{lo} naive phenotype with only a negligible portion (< ~1%) of conversion into CD44^{hi} cells (**provided as the reviewer's perusal in Reviewer Fig. 20**); as expected, a significant fraction (~20%) of CD5^{hi} Ly6C⁻ (but not CD5^{lo} Ly6C⁻) donor cells became Ly6C⁺ cells. Therefore, we conclude that Ly6C⁺ cells are truly naive and not a transitional subset developing toward MP (or VM) cells.

After sorting
(Before transfer)

8 weeks after transfer

Reviewer Figure 20. Naive CD8⁺ T cell subsets (CD5^{lo}, CD5^{hi} Ly6C⁻, CD5^{hi} Ly6C⁺) maintain CD44^{lo} naive phenotype even after 8 weeks. FACS-purified naive CD8⁺ T cell subsets (CD5^{lo}, CD5^{hi} Ly6C⁻, CD5^{hi} Ly6C⁺) were transferred to congenically different B6 hosts. After 8 weeks, CD44 and Ly6C expressions of the donor cells were analyzed by flow cytometry.

3. The dependency of Ly6C expression on type I interferons is an interesting finding, but the novelty is undercut by previous work demonstrating the same effect (e.g. References 16-17).

We feel strongly that this criticism is unwarranted. Our data are indeed in accord with the previous finding that T1IFN leads to the induction of Ly6C on both CD4⁺ and CD8⁺ T cells in vitro and in vivo (which we have also acknowledged in our paper). But the point to stress here is that, beautiful though they are, the in vivo data on Ly6C expression have been tested virtually under strong pro-inflammatory conditions, and thus the physiological relevance of the steady-state role of tonic level of T1IFN, especially with regard to normal T cell homeostasis, is largely unclear. In particular, no-one to our knowledge has shown that differences in the relative self-reactivity on naive CD8⁺ T cells could radically alter the sensitivity of these cells to in vivo T1IFN under steady-state condition – the main point of

our paper. There is much interest in addressing the question of how post-thymic pre-immune T cell populations would shape their phenotypic and functional heterogeneity, but here the emphasis is on CD5 expression, high expression of CD5 being a marker for cells with above-average self-MHC reactivity. But we show in the paper that, at least for naive CD8⁺ T cells, CD5 expression correlates closely with T1IFN (and presumably other homeostatic cues as well) sensitivity, and that it is highly likely that some of the heterogeneity is, at least in part, controlled by T1IFN in a self-driven process. Our overall conclusion therefore is that the intrinsic self-reactivity of different naive CD8⁺ T cell subsets varies considerably and that this difference is controlled largely, if not exclusively, by relative sensitivity to limiting levels of in vivo T1IFN. We suggest this conclusion is indeed novel and of considerable general interest.

Minor Points:

4. In panels such as Figure 1b, analogous plots should also depict marker expression by CD44^{hi} CD8⁺ T cells, as a reference.

According to the reviewer's suggestion, we now have included CD44^{hi} cell markers in comparison with three subsets of CD44^{lo} cells (**new data in Supplementary Fig. 1b in the revised manuscript**).

5. For the RNA-Seq in Figure 3, were cells first gated on CD44^{lo}? This is not mentioned in the text or figure legend.

We are sorry for the shortage of information on the cells used for RNA-seq analysis. As mentioned in the Methods section, all experiments (**previous data in Fig. 3**) were performed with FACS-purified CD44^{lo} naive CD8⁺ T cell subsets. For clarity, we now have added this information in the figure legend of the revised manuscript.

6. Are Ly6C⁺ thymocytes recirculating mature T cells, or T cells that have recently developed?

With regard to the origin of Ly6C⁺ thymocytes, we think that Ly6C⁺ cells seen in the thymus are mostly, if not all, thymus-originated, based on the following two data: 1) Ly6C⁺ (CD44^{lo} CD24^{lo} CD8⁺) cells were clearly detectable in the thymus even day 1 after birth (**previous data in Fig. 1e**) and 2) the thymic Ly6C⁺ cells, unlike peripheral Ly6C⁺ naive cells, failed to show IL-12/IL-18-induced innate response (**previous data in Supplementary Fig. 4b**).

To further confirm the origin of Ly6C⁺ thymocytes, we performed additional experiments using a previously reported "fate-mapping" mouse model, namely Tcrδ^{CreER} R26^{ZsGreen} mice (Zhang et al, PNAS, 2016). As this strain allows for tracking developing T cells via labeling ZsGreen in a time-dependent manner (via tamoxifen-inducible Cre located in *TCRd* gene locus), we can accurately validate thymus-originated Ly6C⁺ cells. As a result, a small fraction of Ly6C⁺ cells (~6-10%) were clearly detectable in ZsGreen⁺ CD8 single-positive (SP) thymocytes during 5-14 days of thymopoiesis (**provided as the reviewer's perusal in Reviewer Fig. 21**). Hence, we conclude that Ly6C⁺ cells seen in the thymus are indeed recently generated thymus-derived subset.

Reviewer Figure 21. Ly6C⁺ cells were detectable in ZsGreen⁺ CD8 single-positive thymocytes during 5-14 days of thymopoiesis. Tamoxifen was injected to TCR δ^{CreER} R26^{ZsGreen} mice at different time points (5-14 days before sacrifice). Thymus were harvested and used to analyze recently developing T cells. Ly6C⁺ cells were analyzed from ZsGreen⁺ CD8 SP thymocytes (tamoxifen induced recently developing T cells).

7. How is the CD5-lo vs. CD5-hi gate drawn? CD5 expression density has a continuous distribution, and gating seems to be made arbitrarily.

Since CD5 expression shows a relatively narrow range of continuous distribution, we sorted naive CD8⁺ T cells based on the high and low level of CD5 expression reflecting the upper and lower ~20% of CD5 distribution, respectively, which have been used in several previous reports (Cho et al, Immunity, 2010; Mandl et al, Immunity, 2013; Persaud et al, Nat. Immunol., 2014; Fulton et al, Nat. Immunol., 2015; Cho et al, Nat. Commun., 2016).

8. In Figure 5a, what are the ratios of donor cell populations at endpoint if mice are not infected with LCMV? Is there a difference in survival and engraftment at baseline?

To address the reviewer's valid question, we performed additional co-transfer experiments in which FACS-purified CD5^{lo} Ly6C⁻, CD5^{hi} Ly6C⁻ and CD5^{hi} Ly6C⁺ B6 naive CD8⁺ T cells were mixed at 1:1 ratio (i.e., CD5^{lo} Ly6C⁻ + CD5^{hi} Ly6C⁻; CD5^{lo} Ly6C⁻ + CD5^{hi} Ly6C⁺; and CD5^{hi} Ly6C⁻ + CD5^{hi} Ly6C⁺), adoptively transferred to congenically distinct B6 hosts, and then relative survival of donor cells co-transferred was analyzed at different time points (1, 4 and 8 weeks post-transfer). All three naive subsets showed similar extent of steady-state survival throughout the entire analysis period, although CD5^{lo} cells (relative to both CD5^{hi} subsets co-transferred) showed slightly enhanced engraftment during the first week after adoptive transfer (provided as the reviewer's perusal in Reviewer Fig. 22). So, we conclude that the difference in LCMV-specific response is not due to differential in vivo survival (and engraftment) of the three naive CD8⁺ T cell subsets.

Reviewer Figure 22. Naive CD8⁺ T cell subsets (CD5^{lo}, CD5^{hi} Ly6C⁻, CD5^{hi} Ly6C⁺) show similar degrees of steady-state survival. FACS-purified naive CD8⁺ T cell subsets (CD5^{lo}, CD5^{hi} Ly6C⁻, CD5^{hi} Ly6C⁺) were acquired from Ly5.1/5.1 and Ly5.1/5.2 B6 mice. Each subset (i.e. Ly5.1/5.1 CD5^{lo}, CD5^{hi} Ly6C⁻) was mixed with congenically different other subset (i.e. Ly5.1/5.2 CD5^{hi} Ly6C⁻ or Ly5.1/5.2 CD5^{hi} Ly6C⁺) at 1:1 ratio, generating 3 different combinations (a) CD5^{lo} + CD5^{hi} Ly6C⁻; (b) CD5^{lo} + CD5^{hi} Ly6C⁺; (c) CD5^{hi} Ly6C⁺ + CD5^{hi} Ly6C⁻, and transferred to Ly5.2/5.2 WT B6 hosts. After 1, 4, or 8 weeks, ratio of surviving donor cells were analyzed for their ratio.

9. A useful positive control for many experiments would be CD44^{hi} CD122⁺ VM cells, to see how these compare to the Ly6C⁺ “naive” cells.

As mentioned above (see our reply to the reviewer’s question 1), we have now provided numbers of additional new data showing that Ly6C⁺ naive cells are distinctly different from CD44^{hi} MP (and VM) cells in their surface phenotype, gene expression profiles and functional properties. Although in this paper we have only focused on comparing naive subsets, number of previous reports have already well demonstrated that MP (and VM) cells have many features in common with antigen-experienced true memory cells and induce much greater immune responses to pathogen infections than naive counterparts (Hamilton et al, Nat Immunol., 2006; Haluszczak et al, J. Exp. Med., 2009; Lee et al, PNAS, 2013). So, we think that adding CD44^{hi} cells is, though useful, beyond the scope of our study focusing on defining phenotypic and functional heterogeneity within naive cells.

10. For Figure 5, what is the natural distribution of naive CD5^{lo}, Ly6C⁻, and Ly6C⁺ populations in P14 Rag1^{-/-} mice?

As P14.*Rag1*^{-/-} mice are a monoclonal TCR transgenic, naive CD8⁺ T cells in these mice, similar to other monoclonal TCR transgenics (e.g., OT-I), show a much narrower distribution of CD5 expression than that of polyclonal B6 counterparts. So, the level of CD5 expression between Ly6C⁻ and Ly6C⁺ cells in P14 naive CD8⁺ T cells is also narrow with ~10-15% of P14 cells being Ly6C⁺ cells (**provided as the reviewer’s perusal in Reviewer Fig. 23**). Therefore, in all of our experiments with P14 naive CD8⁺ T cell subsets, we sorted these cells into CD5^{lo} Ly6C⁻, CD5^{hi} Ly6C⁻ and CD5^{hi} Ly6C⁺ subsets based on the lower ~10% (for CD5^{lo} Ly6C⁻) and the upper ~40-50% (for both CD5^{hi} Ly6C⁻ and CD5^{hi} Ly6C⁺) of CD5 expression distribution, respectively.

Reviewer Figure 23. Proportions of CD5^{lo}, CD5^{hi} Ly6C⁻, and CD5^{hi} Ly6C⁺ cells in naive P14 CD8⁺ T cells. Proportion of naive CD8⁺ T cell subsets were analyzed from splenocytes of P14 Rag1^{-/-} mouse.

Reviewer #3 (Remarks to the Author):

This paper describes the genetic and functional differences between CD5^{hi} Ly6C⁻ and Ly6C⁺ CD8 T cells. They find that development of the latter requires tonic exposure to type I IFNs, and results in cells that expand more rapidly after LCMV infection and are more likely to differentiate into short-lived effectors and less likely to become memory cells. The authors conclude that in addition to some degree of self-reactivity, exposure to steady-state levels of type I IFNs shape the functional response of CD8 T cells. Although the focus of this work is rather narrow, there are a great deal of data, often generated in adoptive transfer models in which the different subsets “compete” in the same environment, that support the conclusions.

Specific comments:

1. In Figs. 5a,b it is shown that Ly6C⁻ cells expand less well in response to LCMV than Ly6C⁺ cells. The authors say this is “surprising” but don’t explain why they think so. I agree it is surprising, because LCMV rapidly induces type I IFN production, and as shown in vitro they induce Ly6C expression on previously Ly6C⁻ cells in at least 24 hrs, perhaps less. So unlike the steady-state experiments, one might have thought that during a viral response with much higher type I IFN levels there would be little difference between cells that were Ly6C⁻ vs. Ly6C⁺ at the time of transfer.

Here, the reviewer pointed out the lack of explanation on why our data showing greater antigen-specific expansion of Ly6C⁺ cells than Ly6C⁻ cells upon LCMV infection is surprising. In this regard, we would like to emphasize the following two explanations. First, at face value, we expected similar response, as these two naive subsets express equivalently high levels (CD5^{hi}) of CD5 expression and previously, CD5^{hi} cells have been well demonstrated to show higher antigen-specific expansion than CD5^{lo} cells (Fulton et al, Nat. Immunol., 2015). In this study, the relatively high self-reactivity of CD5^{hi} cells (together with their enhanced IL-2 responsiveness) has been attributed to explain these phenomena. Therefore, we suggest that the observed difference of CD5^{hi} subsets (i.e., Ly6C⁻ and Ly6C⁺) in our paper is indeed confusing and also surprising, as these findings indicate there is a requirement of additional factors to explain the superior response of Ly6C⁺ cells over Ly6C⁻ cells. Second, in addition to the data with B6 naive subsets, the very similar phenomena also applied to monoclonal P14 cells. We thought that this is an additional surprising point, as the data with P14 subsets indicate that the enhanced LCMV-specific response with Ly6C⁺ cells (relative to Ly6C⁻ cells) is not due to mere correlation reflecting functional diversity among polyclonal B6 repertoire.

With regard to the possible effect of T1IFN produced at higher levels upon LCMV infection we addressed this by performing additional co-transfer experiments. FACS-purified naive CD8⁺ T cell subsets from P14.*Ifnar*^{-/-} (KO) and P14 WT mice were transferred to B6 hosts (1:1 ratio of KO and WT CD5^{lo}, KO and WT CD5^{hi} CD183⁻, or KO and WT CD5^{hi} CD183⁺ cells; note that CD183 was used due to the lack of Ly6C expression on KO cells) and then infected with LCMV (provided as the reviewer’s perusal in Reviewer Fig. 24a). With these cells lacking IFNAR, the antigen-specific expansion (day 7 post-infection), unlike WT P14 donor cells, were substantially reduced in all KO donor cells co-transferred, including both CD5^{hi} CD183⁻ and CD183⁺ cells (provided as the reviewer’s perusal in Reviewer Fig. 24b). Notably, there was also a significant alteration in the fate of effector differentiation,

showing a decrease in SLEC but conversely an increase in MPEC (provided as the reviewer's perusal in Reviewer Fig. 24c, d).

Hence, based on these new data and our previous data (previous data in Fig. 7e-g), we suggest that the effect of T1IFN on T cells is fundamentally different, largely depending on the states of responding T cells (naive vs. antigen-stimulated) and immune contexts (steady-state vs. pathogen infected). For clarity, we thus avoid stating the role of LCMV-induced proinflammatory T1IFN in our paper, as its effect is much broader and stronger across all subsets of naive CD8⁺ T cells responding to pathogen infection.

Reviewer Figure 24. All naive CD8⁺ T cell subsets from P14.Ifnar^{-/-} mice show decreased antigen-specific expansion and have skewed effector differentiation towards MPEC compared to those from P14 WT mice. FACS-purified naive CD8⁺ T cell subsets (CD5^{lo}, CD5^{hi} CD183⁻, CD5^{hi} CD183⁺) from P14 WT and P14.Ifnar^{-/-} (KO) mice were mixed at 1:1 ratio (i.e. WT CD5^{lo} + KO CD5^{lo}; WT CD5^{hi} CD183⁻ + KO CD5^{hi} CD183⁻; WT CD5^{hi} CD183⁺ + KO CD5^{hi} CD183⁺) and transferred to WT B6 hosts, followed by LCMV infection a day after. After 7 days, (b) proportion of donor cells in CD8⁺ T cells and (b) SLEC and (c) MPEC differentiation were analyzed.

2 In panels 2f and 2g, how much IFN β was added? In panel h, what is the unit for the numbers .25, 1, and 5 – concentration of IFN β , or time?

We apologize for the lack of exact information on the experiments pointed by the reviewer and now have corrected them in the revised version of manuscript.

3. In Fig. 2e, there was only a difference at one concentration, 0.1 and 1, so it's not possible to say how different the responses really are with much certainty. A finer titration between those 2 points would be helpful (e.g. 2-fold). Also, units missing on the Y axis – $\mu\text{g/ml}$? μM ? What?

We apologize for insufficient information on the figure and now corrected them in the revised version of manuscript.

With regard to the inaccuracy of our dose response data (Fig. 2e), we performed additional titration experiments with FACS-purified CD5^{lo} and CD5^{hi} Ly6C⁻ cells and now addressed the reviewer's concern. The result clearly show that the effect of IFN- β on Ly6C expression in vitro was much less (~8-fold) pronounced on CD5^{lo} cells than CD5^{hi} cells (0.2 ng/ml vs. 1.6 ng/ml of EC50; provided as the reviewer's perusal in Reviewer Fig. 25a, and also added in Supplementary Fig. 2f in the revised manuscript). Similar results were also observed with P14 CD5^{lo} and CD5^{hi} Ly6C⁻ cells (provided as the reviewer's perusal in Reviewer Fig. 25b).

Reviewer Figure 25. Ly6C induction by T1IFN in CD5^{lo} Ly6C⁻ and CD5^{hi} Ly6C⁻ cells. CD5^{lo} Ly6C⁻ (CD5^{lo}) and CD5^{hi} Ly6C⁻ (CD5^{hi}) naive CD8⁺ T cells were FACS-purified from (a) B6 or (b) P14 mice. Purified cells were cultured with titrated concentrations (0.05 to 1.6 ng/ml; 2-fold increase) of IFN-β overnight. Proportion of Ly6C⁺ cells were analyzed by flow cytometry.

4. What are T1IFN-induced donor-derived Ly6C⁺ cells (inLy6C⁺), I did not see them defined as such? In the text it seems they are generated in vivo (“Ly6C⁻ P14 cells were transferred to and parked in B6 hosts for 7 and 21 days to generate Ly6C⁺ cells intermittently induced by T1IFN (inLy6C⁺)”, but what does “intermittently” then refer to? Are they the cells described in the Methods section “in vitro Ly6C induction”? If so, they should be defined as inLy6C⁺ there, too, so there is no ambiguity.

We apologize for providing ambiguous and rather confusing information on the experimental conditions. As we stated in the paper, the term of inLy6C⁺ cells was to define newly “induced” Ly6C⁺ cells from adoptively transferred Ly6C⁻ cells over a relatively short period of time (~7-21 days). Since Ly6C expression was dependent on a T1IFN signal, newly formed Ly6C⁺ (inLy6C⁺) cells meant that these cells might have been “intermittently” (if not continuously) receiving steady-state tonic T1IFN signals for a short period of time. In the revised manuscript, we have corrected the sentence that may cause some confusion (**Page 16, lines 6-7**), and also added the experimental conditions for generating “inLy6C⁺” cells in the Method section.

5. It seems unlikely, but since the authors have the data: are the TCR levels different on Ly6C⁻ and Ly6C⁺ CD8 T cells?

As the reviewer points out, MFIs for TCR levels on P14 cells are indeed slightly higher in Ly6C⁻ cells than Ly6C⁺ cells (**previous data in Supplementary Fig. 5f**). Since this phenomenon was also observed in the corresponding B6 naive subsets (**provided as the reviewer’s perusal in Reviewer Fig. 26**) and in a previous report (Fulton et al, Nat. Immunol., 2015), we think that slightly lower TCR expression on Ly6C⁺ cells (relative to Ly6C⁻ cells) is not directly relevant for the higher antigen-specific response of Ly6C⁺ cells than Ly6C⁻ cells.

Reviewer Figure 26. Naive CD5^{lo} CD8⁺ T cells slightly higher TCRβ expression compared to naive CD5^{hi} Ly6C⁻ or CD5^{hi} Ly6C⁺ cells. Ex vivo splenocytes of WT B6 were stained for TCRβ and analyzed by flow cytometry.

REVIEWERS' COMMENTS

Reviewer #1 (Remarks to the Author):

The authors have provided extensive data to respond to my initial concerns. While the new data on the loss of IFN responsiveness being secondary to the canonically described TCR-pMHC signals, does dampen the original emphasis, this is discussed in the manuscript. Thank you for taking the time to provide a scholarly response.

Reviewer #2 (Remarks to the Author):

In this revised manuscript, Ju and colleagues have added a number of key pieces of data demonstrating the purity of the 3 naive CD8+ T cell populations under investigation, and the properties of these subsets in comparison with those of memory-phenotype (MP) CD8+ T cells. The results nicely demonstrate that the properties observed for CD5-hi Ly6C+ naive cells are unlikely to be due to significant contamination by MP cells. This is shown by a number of key figure additions, including Supplementary Figures 1b, 1c, 1d, 2e, 3a, 4a as well as Reviewer Figures 14 and 15. The response also includes new data in Reviewer Figures 21 and 22 addressing other points. These experiments provide a rigorous analysis demonstrating that the cells in question are truly naive. Overall, this study makes valuable contributions to our understanding of heterogeneity in the naive T cell pool, and is appropriate for publication in this journal.

Reviewer #3 (Remarks to the Author):

The authors provided new data and satisfactory answers my comments.

Reply to the comments raised by Reviewers:

Reviewer #1 (Remarks to the Author):

The authors have provided extensive data to respond to my initial concerns. While the new data on the loss of IFN responsiveness being secondary to the canonically described TCR-pMHC signals, does dampen the original emphasis, this is discussed in the manuscript. Thank you for taking the time to provide a scholarly response.

Reviewer #2 (Remarks to the Author):

In this revised manuscript, Ju and colleagues have added a number of key pieces of data demonstrating the purity of the 3 naive CD8+ T cell populations under investigation, and the properties of these subsets in comparison with those of memory-phenotype (MP) CD8+ T cells. The results nicely demonstrate that the properties observed for CD5hi Ly6C+ naive cells are unlikely to be due to significant contamination by MP cells. This is shown by a number of key figure additions, including Supplementary Figures 1b, 1c, 1d, 2e, 3a, 4a as well as Reviewer Figures 14 and 15. The response also includes new data in Reviewer Figures 21 and 22 addressing other points. These experiments provide a rigorous analysis demonstrating that the cells in question are truly naive. Overall, this study makes valuable contributions to our understanding of heterogeneity in the naive T cell pool, and is appropriate for publication in this journal.

Reviewer #3 (Remarks to the Author):

The authors provided new data and satisfactory answers my comments.

We appreciate the reviewers' valuable comments and questions and the reviewers agreed that all issues being raised have been thoroughly addressed in the revised manuscript.